# Sulfur-enriched sub-arc fluids drive deep sulfur cycling in subduction zones

Dong-Bo Tan[1], Yilin Xiao [1,2]✉, Yibing Li [3], Haiyang Liu [4], Deshi Jin[1], Yang-Yang Wang[1], Xiaoguang Li [5], Haihao Guo[6], Zeng-Li Guo[7], Carlos J. Garrido [8] & Timothy Kusky [6]

Arc magmas are enriched in sulfur relative to mid-ocean ridge basalts, commonly attributed to slab-derived sulfur inputs during subduction. However, the contribution of slab fluids remains debated because sulfur concentrations in sub-arc fluids have not been directly measured. Here we quantify sulfur in slab-derived fluids preserved as multiphase fluid inclusions composed of $H_2O$, calcite, and chalcopyrite in omphacite from ultrahigh-pressure eclogites in the Sumdo orogenic belt. Three-dimensional Raman spectroscopy reveals high sulfur concentrations averaging ~6 wt.%. Mass-balance calculations indicate that such fluids can efficiently enrich the mantle wedge and supply up to ~70% of the sulfur emitted by arc volcanism. We further suggest that chalcopyrite formed through post-entrapment reduction of oxidized sulfur species by host omphacite, followed by precipitation with co-entrapped copper and iron. Our findings identify sub-arc depths as a critical window for slab sulfur release and provide key constraints on deep sulfur cycling and copper mobilization in arc systems.

Sulfur is a key volatile element that profoundly influences a wide range of Earth system processes, including volcanic outgassing, ore genesis, climate regulation, and biological evolution[1–3]. Subduction zones represent the primary pathway for sulfur exchange between Earth's surface and interior, transferring sulfur from the subducting slab into the mantle[4,5]. However, the ultimate fate of subducted sulfur remains uncertain, particularly the extent to which it is transferred into the mantle wedge to sustain arc-related sulfur outputs versus retained in the slab and recycled into the deep mantle[6–10].

Arc volcanic rocks are characteristically enriched in sulfur, a feature widely attributed to sulfur input from slab-derived fluids[11–21]. However, mass-balance calculations indicate that such fluids contribute a highly variable fraction of arc-related sulfur emissions, revealing a mismatch between the observed arc outputs and the estimated slab-derived inputs[18–22]. For example, sulfur isotopic compositions of ultrahigh-pressure (UHP) metamorphic rocks, combined with thermodynamic modeling, suggest that slab-derived fluids may account for only ~20% of the sulfur budget of arc magmas[22,23]. In contrast, sulfur enrichments recorded in melt inclusions from natural arc basalts imply substantially larger contributions of ~40–70%[20]. This discrepancy may partly reflect the compositional diversity of slab-derived fluids among different subduction systems. However, it may also arise from the lack of direct constraints on sulfur concentrations and fluxes in natural slab-derived fluids themselves. Resolving this uncertainty requires direct quantification of sulfur in slab-derived fluids at sub-arc depths, where fluid-mediated

[1]State Key Laboratory of Lithospheric and Environmental Coevolution, School of Earth and Space Sciences, University of Science and Technology of China, Hefei, China. [2]Chinese Academy of Sciences Center of Excellence in Comparative Planetology, Hefei, China. [3]Marine Science and Technology College, Zhejiang Ocean University, Zhoushan, China. [4]Key Laboratory of Ocean Observation and Forecasting, Center of Deep Sea Research, Chinese Academy of Sciences, Qingdao, China. [5]State Key Laboratory of Lithospheric and Environmental Coevolution, Institute of Geology and Geophysics, Chinese Academy of Sciences, Beijing, China. [6]State Key Laboratory of Geological Processes and Mineral Resources, School of Earth and Planetary Sciences, China University of Geosciences, Wuhan, China. [7]SK Lab-DeepMinE, MOE KLab-OBCE, School of Earth and Space Sciences, Peking University, Beijing, China. [8]Instituto Andaluz de Ciencias de la Tierra (IACT–CSIC), CSIC, Granada, Spain. ✉e-mail: ylxiao@ustc.edu.cn

sulfur transfer exerts the strongest influence on the overlying mantle wedge.

To date, estimates of sulfur concentrations in subduction zone fluids have relied predominantly on indirect approaches, including analyses of sulfur concentrations and isotopic compositions in exhumed metamorphic rocks, melt inclusions in arc basalts and tephra, high-temperature and high-pressure (HT–HP) experiments, and thermodynamic modeling[18–22,24–29]. While these approaches have provided valuable insights, they are subject to substantial uncertainties related to pressure–temperature–redox conditions, phase relations, and fluid–rock equilibria. As a result, reported sulfur concentrations in slab-derived fluids span a wide range, from <1 wt% to several weight percent[22,26–29]. For example, thermodynamic calculations based on the deep Earth water (DEW) model[22] predict sulfur concentrations of ~0.5–1.0 wt%, which may be underestimated owing to extrapolation beyond the model's calibration range. Experimental studies[24,25] have reported sulfur contents of 1.58–14.83 wt% in sulfide-bearing basaltic systems, but these experiments were conducted at temperatures higher than those characteristic of globally dominant cold subduction zones and employed simplified starting compositions that may not fully capture the complexity of natural fluid–rock interactions. Crucially, direct measurements of sulfur concentrations in natural sub-arc fluids have not yet been reported, remaining a major barrier to quantifying slab-to-mantle sulfur transfer.

A more direct approach to constraining slab-derived fluid compositions involves the study of fluid inclusions preserved in HP–UHP metamorphic rocks[30,31]. Among the major sulfur reservoirs entering subduction zones—altered oceanic crust, marine sediments, and serpentinized peridotite—the altered oceanic crust typically dominates sulfur input in oceanic subduction systems[18,21]. During prograde metamorphism and dehydration, fluids released from this source may be trapped in co-crystallizing metamorphic minerals, thereby preserving snapshots of fluid compositions at the time of entrapment, commonly with limited post-entrapment modification[32]. In particular, fluid inclusions hosted in omphacite formed under eclogite-facies conditions are especially valuable, as they provide rare access to the chemistry of slab-derived fluids at sub-arc depths. Despite this potential, however, direct quantitative constraints on sulfur concentrations in such fluid inclusions remain scarce.

Here, we present in situ sulfur measurements from chalcopyrite-bearing multiphase fluid inclusions hosted in omphacite from UHP eclogites in the Sumdo orogenic belt, southern Tibet. These inclusions preserve evidence of fluids released during the dehydration of altered oceanic crust at sub-arc depths. High-resolution three-dimensional (3D) Raman spectroscopy reveals average sulfur concentrations of ~6 wt%, providing the direct quantification of sulfur in natural subduction zone fluids. Our results establish sub-arc depths as a critical window for sulfur mobilization and provide key insights into sulfur cycling between the subducting slab, mantle wedge, and arc outputs. Furthermore, we highlight the role of sulfur-rich oxidized sub-arc fluids in facilitating efficient copper mobilization, with important implications for the formation of porphyry copper deposits in arc settings.

## Results

### Geological setting and multiphase fluid inclusions
The Sumdo eclogite belt, located in the east-central Lhasa terrane of the Tibetan Plateau, represents a well-preserved segment of a fossil subduction zone formed during the northward subduction of the Paleo-Tethys Ocean beneath the North Lhasa terrane[33] (Fig. S1). The belt extends over 100 km in an east–west direction and is 5–10 km wide. It comprises a diverse lithological assemblage including eclogite, blueschist, greenschist, marble, mica schist, quartzite, and volcanic rocks[33–35] (Fig. S1). The coexistence of eclogite- and blueschist-facies rocks points to a cold subduction regime characterized by a low geothermal gradient. Peak metamorphic conditions recorded by eclogites reach ~2.6–2.7 GPa and 630–780 °C[33], corresponding to sub-arc depths in a typical oceanic subduction setting[36]. The occurrence of coesite pseudomorphs in garnet and omphacite further attests to UHP metamorphism[37]. The investigated eclogites are composed primarily of garnet and omphacite, with minor rutile, phengite, and amphibole, and sparse sulfide minerals—mainly pyrite[35](Fig. S2). Garnet grains display well-developed compositional zoning, recording crystal growth from prograde metamorphism through peak conditions and into the early retrograde stage[35,37]. In contrast, omphacite is compositionally unzoned and occurs both as porphyroblasts and as inclusions within garnet, consistent with crystallization during a single metamorphic stage—most likely at peak conditions[37].

Fluid inclusions occur in both garnet and omphacite, but their abundance, distribution, and preservation differ markedly between the two host minerals. Inclusions in garnet are generally sparse and commonly show evidence of modification during retrograde metamorphism. In contrast, omphacite hosts abundant, well-preserved inclusions that are spatially restricted to crystal cores. These omphacite-hosted inclusions display several petrographic features, as outlined below. First, the fluid inclusions are confined to the cores of omphacite grains and do not form trails that crosscut the host crystals (Fig. 1), as would be expected for secondary inclusions trapped along healed fractures. Detailed petrographic observations further reveal no microfractures or healed cracks in the host omphacite that could have acted as pathways for late-stage fluid infiltration (Fig. S2). Second, the fluid inclusions commonly occur as isolated or as arrays parallel to the crystallographic c-axis of omphacite (Fig. 1). Individual fluid inclusions are typically ~5–60 μm in length and ~2–8 μm in width and display a consistent columnar morphology that mimics the elongated crystallographic habit of the host mineral. This negative-crystal-like geometry is consistent with primary entrapment and provides supporting, though not diagnostic, evidence for a primary origin[38]. Third, the fluid inclusions show uniform filling degrees of ~80–90% at room temperature and contain consistent proportions of liquid, solid, and vapor phases (Fig. 1), a feature inconsistent with heterogeneous secondary trapping. Combined with the fact that omphacite crystallized exclusively at the peak stage of UHP conditions[37], corresponding to sub-arc depths[36], these observations suggest that omphacite-hosted fluid inclusions provide the most reliable recorders of slab-derived fluid compositions at sub-arc conditions.

### Raman spectroscopic characterization and multiphase fluid composition reconstruction
High-resolution Raman spectroscopy analysis of all well-preserved primary multiphase fluid inclusions reveals a consistent three-phase assemblage comprising chalcopyrite, calcite, and aqueous fluid (Figs. 2, S3, S4, and see "Methods"). The volumetric proportions of solid and liquid phases are remarkably uniform across inclusions (Supplementary Data 1 and Fig. S5), indicating entrapment of a compositionally homogeneous fluid. To quantitatively reconstruct the bulk composition of the trapped fluids, we applied three-dimensional (3D) Raman mapping to quantify the relative volumes of each phase, followed by mass-balance calculations using published density and compositional data for chalcopyrite, calcite, and brine (Supplementary Data 1 and see "Methods"). The reconstructed fluid compositions show a high degree of consistency among inclusions (Fig. S5), yielding an average concentration of 6.33 wt% sulfur, 6.33 wt% copper, 5.54 wt% iron, 10.16 wt% CaO, 15.23 wt% carbon (as $CO_3^{2-}$), 3.22 wt% sodium, 5.25 wt% chlorine, and 47.96 wt% $H_2O$ (Supplementary Data 2).

## Discussion

### Sulfur content in fluids derived from altered oceanic crust
In situ analyses of multiphase fluid inclusions hosted in omphacite from UHP eclogites in the Sumdo orogenic belt reveal high sulfur

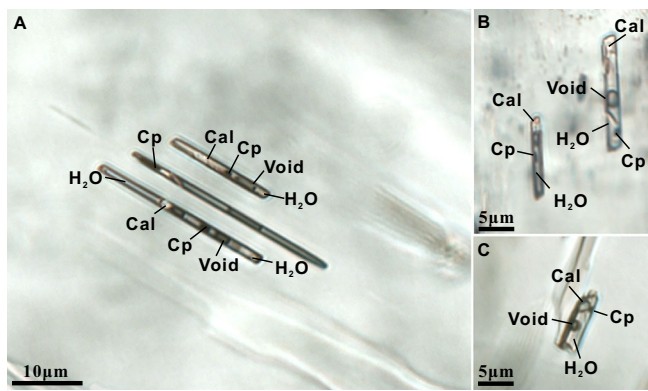

**Fig. 1 | Photomicrographs of multiphase fluid inclusions hosted in omphacite from the Sumdo eclogite, southern Tibet. A–C** Inclusions consistently comprise $H_2O$, chalcopyrite (Cp), calcite (Cal), and a vapor phase (void).

concentrations, with an average of ~6 wt%. Multiple independent lines of evidence indicate that these fluid inclusions faithfully record the composition of slab-derived fluids released during eclogite-facies dehydration at sub-arc depths. First, as discussed above, the investigated inclusions were trapped during omphacite crystallization at peak metamorphic conditions. Their confinement to omphacite cores, together with uniform petrographic characteristics (Fig. 1), supports preservation of slab-derived fluid compositions under UHP conditions. Consistent with this interpretation, ternary diagrams constructed from both phase proportions and reconstructed bulk compositions display tight clustering among fluid inclusions (Fig. S5), demonstrating a high degree of compositional uniformity and supporting derivation from a common fluid source. Second, chalcopyrite is the only sulfide phase observed within the fluid inclusions, whereas pyrite dominates the sulfide assemblage of the host eclogite (Figs. 1 and S2). This systematic mineralogical contrast argues against mechanical incorporation of sulfides from the host rock and instead indicates in situ precipitation from sulfur-rich trapped fluids. The ubiquitous association of chalcopyrite with calcite and an aqueous phase in all inclusions further underscores the sulfur-rich character of these fluids (Fig. 2). Third, we explicitly assessed the potential effects of post-entrapment interaction between the trapped fluids and host omphacite. A representative multiphase fluid inclusion was exposed by polishing, and three electron probe microanalysis (EPMA) transects were conducted across the surrounding omphacite (see "Methods"). These analyses reveal only minor spatial variations in major-element concentrations, with FeO contents adjacent to the inclusion lower by ~0.4 wt% and CaO differing by ~0.6 wt% relative to regions farther from the inclusion (Fig. S6 and Supplementary Data 3). This pattern suggests localized post-entrapment element exchange. However, the overall extent of modification appears negligible for elements other than Ca and Fe and is unlikely to have substantially altered the bulk composition of the trapped fluids. Consequently, the reconstructed sulfur concentrations remain robust and consistently high. Finally, the geological context of the Sumdo eclogites further supports a slab-derived origin for these sulfur-rich fluids[37,39]. Whole-rock major- and trace-element systematics and Sr–Nd isotopic compositions indicate that the eclogite protoliths were derived predominantly from altered oceanic crust, with contributions from relatively fresh igneous components[33–35,37,39]. In addition, equilibrium oxygen isotope fractionation between quartz and garnet/omphacite further suggests a closed-system fluid regime during eclogite-facies metamorphism, minimizing the influence of externally derived fluids[39]. Based on independently constrained P–T conditions (~2.6–2.7 GPa and 630–780 °C)[33], fluid entrapment occurred at peak metamorphism[36], corresponding to sub-arc depths in a cold subduction setting. Taken together, these observations provide

the direct, in situ quantitative evidence that fluids released during dehydration of altered oceanic crust at sub-arc depths are capable of transporting substantial amounts of sulfur.

Furthermore, the sulfur concentrations measured in this study (~3.2–10.3 wt%, Supplementary Data 2) fall within the broader range (~0.5–15 wt%) inferred for subduction zone fluids from thermodynamic models and high-pressure experiments[22,24–29]. Our measurements of multiphase fluid inclusions therefore validate previous experimental and theoretical predictions of elevated sulfur contents in sub-arc fluids while providing tighter natural constraints on the range of sulfur concentrations in slab-derived fluids. Importantly, sulfur-rich slab-derived fluids are unlikely to be unique to the Sumdo system. Numerous studies based on observations of arc magmas and melt inclusions, together with experimental constraints and thermodynamic modeling, have demonstrated extensive sulfur transfer from the subducting slab to the mantle wedge, highlighting the critical role of slab-derived fluids[11–21,27]. Moreover, sulfide- and sulfate-bearing fluid inclusions have been documented in HP–UHP eclogites and associated veins from other orogenic belts worldwide, such as the western Alps and the Caledonides.[32,40,41,42] Collectively, these observations indicate that sulfur enrichment in slab-derived fluids at sub-arc depths is a common feature of deep subduction zones rather than an isolated characteristic of the Sumdo system.

## Formation of sulfur-rich fluids at sub-arc depths

The high sulfur concentrations recorded in multiphase fluid inclusions from the Sumdo eclogite indicate highly efficient sulfur mobilization during subduction. In altered oceanic crust, sulfur is initially hosted in both sulfate- and sulfide-phases, which are progressively destabilized with increasing depth during subduction[43,44]. Some studies suggest that sulfate-bearing minerals mostly decompose and release oxidized sulfur species ($SO_4^{2-}$) at relatively shallow forearc depths (<70 km), whereas sulfide phases remain stable to greater depths and break down under eclogite-facies conditions (>70 km), liberating reduced sulfur (e.g., $S^{2-}$, $HS^-$) into coexisting fluids[22,23,43,45]. However, more recent experimental, thermodynamic, and geochemical studies indicate that oxidized sulfur species may remain dominant even at sub-arc depths, despite increasing pressure and temperature[18–21,26,27].

To evaluate sulfur speciation in slab-derived fluids and to constrain the origin of chalcopyrite observed within the multiphase fluid inclusions from the Sumdo eclogites, we performed thermodynamic calculations constrained by the geological context and P–T conditions of the Sumdo belt (see "Methods"). The results indicate that slab-derived fluids at sub-arc depths were dominated by oxidized sulfur species, principally $SO_4^{2-}$ and $HSO_4^-$ (Fig. S7), implying that sulfur was predominantly present in an oxidized form at the time of fluid entrapment. This observation is consistent with previous experimental and thermodynamic studies demonstrating that oxidized sulfur species remain highly mobile under deep subduction conditions[26,27]. We further suggest that the sulfur speciation recorded in the multiphase fluid inclusions may have been locally modified after entrapment. As shown in Fig. S6, host omphacite adjacent to exposed fluid inclusions exhibits slightly but systematically lower Fe contents compared with regions farther from the inclusions. This compositional gradient is consistent with limited post-entrapment Fe exchange between the trapped fluids and $Fe^{3+}$-bearing host mineral, potentially driving localized redox reactions. Such interactions could have partially reduced originally oxidized sulfur species and promoted the precipitation of chalcopyrite through reaction with co-entrapped copper and iron within the closed inclusion system. Alternatively, hydrogen diffusion across the inclusion-host boundary may also have induced post-entrapment redox re-equilibration[46,47], potentially resulting in the partial reduction of sulfur within the multiphase fluid inclusions, although the magnitude of this effect is difficult to quantify. Importantly, recent experimental and thermodynamic studies demonstrate

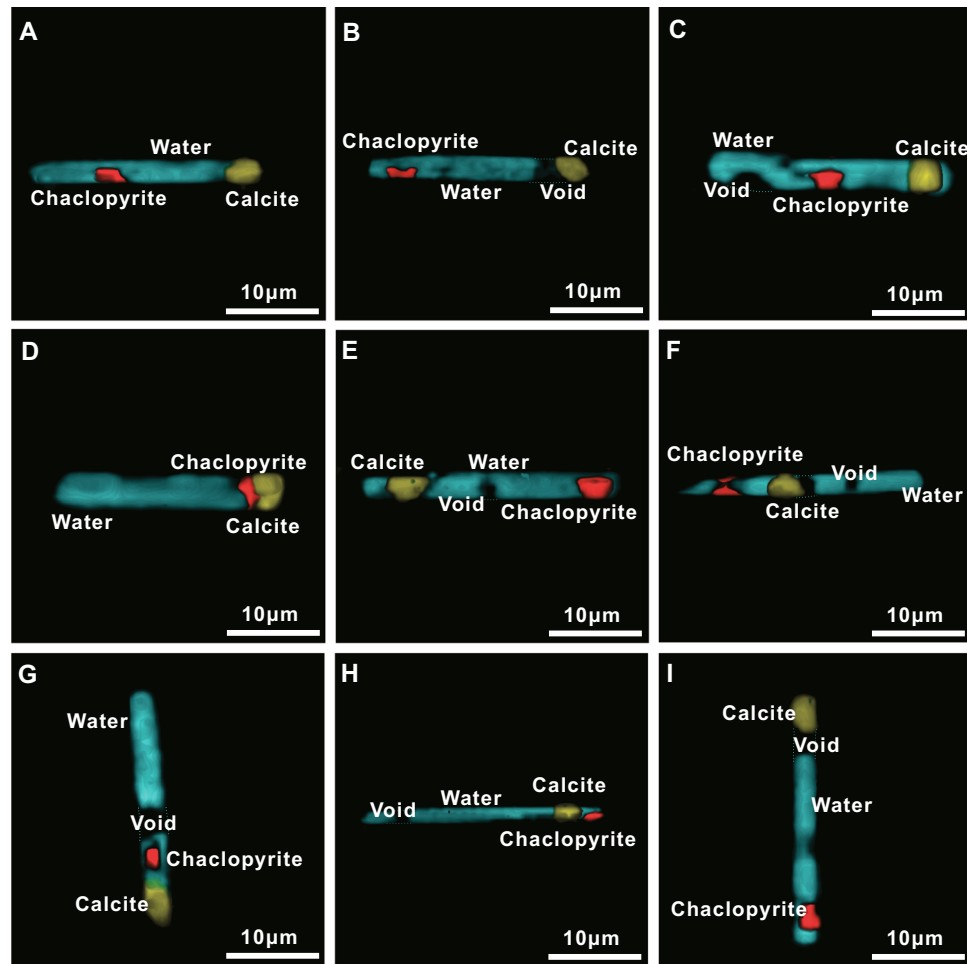

**Fig. 2 | Three-dimensional reconstructions of multiphase fluid inclusions hosted in omphacite from the Sumdo eclogite, southern Tibet. A–I** 3D Raman imaging reveals the consistent internal structure of inclusions, showing coexisting chalcopyrite, calcite, and $H_2O$ within the negative crystal-shaped cavities. These inclusions record the compositions of slab-derived fluids trapped during peak ultrahigh-pressure metamorphism at sub-arc depths.

that oxidized sulfur species can be generated during pyrite–fluid equilibration at pressures of 2–3 GPa and temperatures of 400–800 °C across a wide range of oxygen fugacities[27]. This mechanism is consistent with observations from the Sumdo eclogites, including the presence of residual pyrite in the host rocks (Fig. S2) and the close correspondence between the inferred sulfur-rich oxidized fluid entrapment conditions (~2.6–2.7 GPa, 630–780 °C)[33]. Taken together, these lines of evidence indicate that slab-derived fluids at sub-arc depths primarily transported oxidized sulfur species. Notably, even if localized post-entrapment reduction occurred within individual inclusions, such processes would modify sulfur speciation rather than total sulfur abundance, and therefore do not affect the robustness of the high sulfur characteristics recorded in the fluid inclusions.

Sulfur released from subducting slabs may be transported through a spectrum of liquid media—including aqueous fluids, hydrous melts, and supercritical fluids—whose dominance evolves with increasing depth[48]. Among these, supercritical and near-supercritical fluids formed under HP and HT conditions exhibit distinct physicochemical properties that allow efficient transport of solute-rich components, far exceeding the capacity of conventional aqueous fluids or hydrous melts[30,49]. Multiphase fluid inclusion analyses from the Sumdo eclogites revealed average compositions of ~48 wt% $H_2O$ plus ~52 wt% solutes (Supplementary Data 2), consistent with the solute-rich signatures (30–70 wt%) previously reported for natural supercritical fluids in subduction zones[48]. The estimated

trapped conditions at Sumdo (~2.6–2.7 GPa, 630–780 °C[33]) lie slightly below the second critical endpoint of mafic rock–$H_2O$ systems (~3.0 GPa[50]). However, natural subduction zone fluids are multicomponent systems, and additional constituents—particularly carbon—are expected to modify phase relations. Previous studies suggest that carbonate components can lower the effective P–T threshold for the generation of supercritical fluids[51,52]. In the present study, the presence of abundant dissolved carbon species in the trapped fluids (~3 wt% carbon, dominantly as $CO_2$, $HCO_3^-$, and $CO_3^{2-}$, Fig. S7 and Supplementary Data 2) likely reduced the effective critical conditions, facilitating the formation of near-supercritical, solute-rich fluids at sub-arc depths. Through this mechanism, carbon may indirectly enhance sulfur enrichment by stabilizing high-solubility fluid phases. Taken together, the multiphase fluid inclusions from the Sumdo eclogites most plausibly record entrapped near-supercritical, solute-rich fluids at sub-arc depths. Such fluids are increasingly recognized as highly efficient carriers of slab-derived sulfur. Thermodynamic modeling by Li et al.[22] predicts that peak sulfur release from subducting slabs occurs at depths of ~70–100 km, where fluids may acquire supercritical or near-supercritical properties. The inferred entrapment depth of the Sumdo inclusions (~90 km) falls within this predicted window and lies immediately beneath the zone of arc magma generation. Consistent with this interpretation, recent studies of multiphase fluid inclusions from UHP metamorphic veins in the Dabie orogenic belt, a classic continental subduction zone in China, document elevated sulfur

concentrations (~3 wt%) in supercritical fluids derived from deeply subducted continental crust within the coesite stability field[30], highlighting the high sulfur transport capacity of such fluids. Notably, the even higher sulfur concentrations observed in the Sumdo fluids likely reflect differences in protolith composition[53]—altered oceanic crust (~1000–2000 ppm sulfur) versus continental crust (~400 ppm sulfur)—as well as variations in subduction zone thermal regimes that influence the extent of sulfur liberation at sub-arc depths. Collectively, these observations indicate that near-supercritical, solute-rich fluids at sub-arc depths represent an efficient and widespread mechanism for mobilizing large quantities of sulfur during subduction, with the Sumdo fluid inclusions preserving a direct snapshot of such high-capacity sulfur carriers.

Moreover, recent experimental studies highlight the role of major cations, particularly calcium and sodium, which exert strong and quantifiable controls on sulfate solubility and sulfur transport under UHP conditions[27]. This interpretation is consistent with the elevated CaO (10.16 wt%) and Na (3.22 wt%) concentrations observed in the Sumdo multiphase fluid inclusions, coupled with high sulfur contents (Supplementary Data 2). In addition, we acknowledge that the absolute concentrations of calcium and sodium may be slightly influenced by minor exchange with the host omphacite and by uncertainties associated with estimating fluid salinity using an average value. However, such effects do not alter the overall characterization of the fluids as being enriched in calcium and sodium. These observations therefore support the view that sulfur solubility at sub-arc depths is strongly modulated by fluid major-element chemistry of slab-derived fluids. Similarly, experimental studies have also shown a positive correlation between chlorine concentration and $CaSO_4$ solubility at relatively low pressures (<0.5 GPa), suggesting that chlorine may enhance oxidized sulfur transport under certain conditions[54]. In addition, in subduction environments, the breakdown of sulfides and the progressive decomposition of silicate minerals can liberate both chlorine and iron into coexisting fluids. Because iron exhibits a stronger affinity for chlorine than for sulfur, iron–chlorine complexation may preferentially stabilize iron in solution, potentially suppressing sulfide reprecipitation during fluid migration and thereby facilitating sulfur retention in the fluid phase. Consistent with this scenario, the Sumdo fluid inclusions exhibit elevated concentrations of chlorine (3.5–6.4 wt%), iron (2.80–8.98 wt%), and sulfur (Supplementary Data 2). Nevertheless, we also note that whether the coupled enrichment of chlorine and sulfur reflects similar fluid-mobility behavior under sub-arc conditions, or a direct role of chlorine in enhancing sulfur transport, remains uncertain and requires further investigation. We therefore treat chlorine as a potential but presently not fully constrained secondary contributor to sulfur enrichment in slab-derived fluids.

Collectively, these observations indicate that the elevated sulfur contents preserved in the Sumdo multiphase fluid inclusions reflect the combined effects of several interrelated processes: (1) progressive destabilization of pyrite during dehydration of altered oceanic crust under UHP conditions; (2) generation of near-supercritical, solute-rich fluids at sub-arc depths; and (3) elevated concentrations of major components, including calcium, sodium, chlorine, and dissolved carbon, that enhance sulfur solubility and facilitate its retention in the fluid phase. Together, these processes define an efficient and previously underappreciated mechanism for sulfur mobilization at sub-arc depths, providing critical insights into the dynamics of deep sulfur cycling in subduction zones.

## Implications for the deep sulfur cycling in subduction zones

Previous sulfur mass-balance estimates for subduction zones indicate that the fate of subducted sulfur is highly variable, with sulfur either being transferred into the mantle wedge or retained within the slab and transported into the deeper mantle[6,8,10,18–20,26]. For example, studies of Central American arc basalts suggest that ~10 to 60% of

slab-derived sulfur may be incorporated into the mantle wedge[18]. Such variability likely reflects both heterogeneity in slab-derived fluids and the reliance on indirect constraints in existing mass-balance models. A major source of uncertainty in these estimates is the lack of direct measurements of sulfur concentrations in slab-derived fluids at sub-arc depths, where fluid-mediated sulfur transfer from the slab to the mantle wedge is expected to be most efficient. The multiphase fluid inclusions documented here in the Sumdo eclogites provide the direct, quantitative constraints on sulfur concentrations in fluids released from dehydrating altered oceanic crust under sub-arc conditions. As such, they provide a critical benchmark for evaluating sulfur transfer efficiencies and refining models of deep sulfur cycling in subduction zones.

Although subduction zones span a broad thermal spectrum from cold to hot endmembers, cold subduction zones represent the dominant global regime[36]. In unusually hot subduction systems, slab-derived melts have been proposed as important sulfur carriers, with sulfur concentrations reaching several thousand ppm[55–59]. However, these concentrations remain substantially lower than those measured in slab-derived fluids in this study. More importantly, because most subduction systems worldwide are characterized by relatively low thermal gradients, such as the Sumdo system, our data provide a reasonable basis for extrapolating sulfur fluxes from Sumdo to global cold subduction zones. We acknowledge that sulfur concentrations in slab-derived fluids may vary among different subduction zones owing to differences in slab composition and thermal structure. Nevertheless, the sulfur concentrations determined here (3.2–10.3 wt%, average $6.3 \pm 1.8$ wt%, Supplementary Data 2) fall within the range inferred from previous experimental and thermodynamic studies[22,24–29], but define a more restricted and directly constrained subset. Moreover, the relatively wide range of sulfur concentrations captured by our measurements likely encompasses much of the natural variability expected among global cold subduction systems. Accordingly, to provide a general and conservative assessment of sulfur cycling in oceanic subduction zones at the global scale, we use the full range of measured sulfur concentrations (minimum, maximum, and average ± standard deviation) reported in this study as a key parameter to estimate the global sulfur flux transported by slab-derived fluids at sub-arc depths in cold subduction zones. To quantify this flux, we first constrain the associated $H_2O$ flux (see "Methods"). Based on dehydration modeling constrained by the geological and P–T conditions of the Sumdo system (2.6–2.7 GPa and 630–780 °C[33]), we estimate that approximately 60% of the water initially stored in altered oceanic crust is released at sub-arc depths (Fig. S7 and Supplementary Data 5). Furthermore, treating the Sumdo system as representative of altered oceanic crust dehydration in cold subduction zones, and integrating this dehydration fraction (~60%) with average $H_2O$ contents of altered oceanic crust, global subduction zone length, and altered crustal thickness[60–62], we calculate a global $H_2O$ release flux of ~84.5 Mt/yr from altered oceanic crust at sub-arc depths (~80–90 km). This value closely matches previous independent global estimates (~91 Mt/yr[22] at ~90 km depth), supporting the robustness of our water-flux parameterization. Accordingly, combining the sulfur concentrations measured in this study (3.2–10.2 wt%, average $6.3 \pm 1.8$ wt%) with the estimated $H_2O$ flux (84.5 Mt/yr) yields an estimated global sulfur flux carried by sub-arc fluids of 5.2–21.9 Mt/yr, with an average of ~11.7 Mt/yr (see "Methods"). To further evaluate the significance of this flux, we compare it with global sulfur input flux carried by subducted altered oceanic crust, estimated at ~48.09 Mt/yr[4]. This comparison indicates that approximately $25 \pm 7\%$ of the subducted sulfur is released into fluids at sub-arc depths (see "Methods"). This desulfurization efficiency is consistent with previous model-based predictions[8,26], such as the 25–45% sulfur loss predicted by Walters et al.[26], supporting the reliability of our calculations and underscoring the effectiveness of slab dehydration in mobilizing sulfur at sub-arc

depths. Once released from the slab, these sulfur-rich fluids are capable of significantly enriching the overlying mantle wedge. Mass-balance calculations show that the incorporation of only ~0.2–0.6% of such fluids into the ambient mantle (with a background sulfur content of ~150 ppm) is sufficient to elevate sulfur concentrations to 250–500 ppm—comparable to values observed in metasomatized sub-arc mantle reservoirs[12,63] (Fig. 3 and see "Methods"). This required fluid proportion is lower than previous estimates (~5%) based on trace element and isotopic compositions of arc magmas[64]. We also assessed the timescale required for such sulfur enrichment. Based on the calculated sulfur flux, increasing the sulfur content of the sub-arc mantle from 150 ppm to 250–500 ppm would require approximately 6.5–22.7 Myr (Fig. 3 and see "Methods"). This timescale is broadly consistent with the typical onset of arc magmatism, which initiates within ~1–33 Myr following the onset of subduction[29,65]. Together, the reasonable estimates of both the required fluid addition proportions and the enrichment timescales highlight the critical role of sulfur-rich slab-derived fluids at sub-arc depths. Ultimately, the sulfur stored in the metasomatized sub-arc mantle is returned to the surface via arc volcanism. Comparison between the slab-derived sulfur flux estimated here and global arc volcanic sulfur emissions (~16.8 Mt/yr[4]) indicates that, on average, $70 \pm 20\%$ of sulfur released during arc magmatism can be attributed to slab-derived fluid addition (see "Methods"). These estimates agree well with independent estimates (40–70%) based on melt inclusion data from arc volcanic rocks[18–20], further supporting the reliability of our constraints on slab-derived sulfur input. Collectively, these results highlight the central role of slab-derived fluids at sub-arc depths in regulating sulfur transfer through subduction zones and help reconcile the long-standing imbalance between sulfur inputs and outputs in arc systems (Fig. 4). Moreover, our study establishes a critical observational bridge linking experimental constraints, thermodynamic models, and sulfur-rich signatures inferred from arc magmas.

Furthermore, our results also have important implications for copper cycling in subduction zones. Sulfur acts as a pervasive anionic ligand in deep fluids, strongly enhancing the complexation and transport of chalcophile metals, such as copper[2,28,66–68]. Although some studies argue that copper is largely retained within the slab and is not efficiently transferred by slab-derived fluids[9,69,70], other work suggests that copper can be mobilized under specific conditions, particularly when complexed by sulfur-bearing fluids[71,72]. Our fluid inclusion data provide direct natural evidence that slab-derived fluids at sub-arc depths can contain elevated copper concentrations. While such high copper contents may not be globally ubiquitous and likely reflect localized physicochemical conditions, they nevertheless demonstrate the intrinsic capacity of sulfur-rich, oxidized sub-arc fluids to mobilize and transport substantial amounts of copper. Moreover, geochemical signatures of many arc basalts indicate that their mantle sources are not systematically enriched in copper, implying limited net addition of slab-derived copper to the arc magma source[19,73–75]. In contrast, some metasomatized sub-arc xenoliths show elevated copper compared to the depleted mantle compositions[76–79]. To reconcile this apparent mismatch, we propose three possible processes. First, sulfur may play a dual role in metal cycling, acting both as a facilitator of aqueous copper transport and as a co-precipitant through sulfide saturation during fluid–rock interaction. Under such conditions, copper carried by slab-derived fluids may be efficiently sequestered or filtered at the slab–mantle interface during metasomatic interaction with mantle peridotite, for example, through sulfide precipitation triggered by redox or compositional changes in infiltrating fluids. Second, copper transport may be spatially heterogeneous, with focused fluid flow along channelized pathways leading to localized copper enrichment within the mantle wedge rather than uniform redistribution. Third, sulfide saturation and differentiation processes during magma evolution may buffer copper concentrations in arc melts, thereby reducing compositional contrasts between mid-ocean ridge basalts and arc

lavas[72]. In such cases, variations in slab-derived copper input may be partially masked during magma ascent and evolution. Taken together, these considerations indicate that copper enrichment in arc magmas and the formation of copper deposits are governed by a combination of factors, including the capacity of slab-derived fluids to mobilize copper, the influence of sulfide sequestration during fluid transport, and subsequent magmatic differentiation processes. Nevertheless, our results demonstrate that sulfur-rich, oxidized sub-arc fluids possess the intrinsic ability to efficiently extract and transport copper from the subducted slab. This provides a necessary material foundation for copper enrichment in arc systems and underscores the potential role of sub-arc sulfur-rich, oxidized fluids in the genesis of porphyry copper deposits.

## Methods

### 3D imaging and spectral acquisition

Three-dimensional (3D) Raman mapping of multiphase fluid inclusions hosted in omphacite was conducted at the State Key Laboratory of Lithospheric and Environmental Coevolution, Institute of Geology and Geophysics, Chinese Academy of Sciences, Beijing, China. Analyses were performed using a WITec alpha300R confocal Raman microscope equipped with a 488 nm cobalt laser. The laser power at the sample surface was maintained at 25 nW, which is sufficiently low to minimize laser-induced heating or damage to the inclusions while still ensuring robust detection of Raman signals from both the daughter minerals and associated fluid phase. Spectra were collected over the range of 100–4400 cm⁻¹ to fully capture the Raman features of the multiphase assemblages. Mineral phases were identified and discriminated from compositionally similar counterparts based on established Raman spectral databases (https://rruff.info/) and previous studies (e.g., Frezzotti et al.[80]). For sulfide phases, chalcopyrite (CuFeS$_2$) is characterized by a strong Raman peak at 293 cm⁻¹, whereas other common sulfides display distinct and nonoverlapping diagnostic peaks, including pyrite (FeS$_2$) at 428 cm⁻¹, marcasite (FeS$_2$) at 324 cm⁻¹, sphalerite (ZnS) at 349 cm⁻¹, and galena (PbS) at 136 cm⁻¹. In this study, the dominant sulfide Raman peak consistently occurs at 292–293 cm⁻¹ (Fig. S3), with no additional peaks indicative of other sulfide species, supporting the identification of chalcopyrite as the sole sulfide phase within the inclusions. For carbonate phases, calcite is characterized by strong Raman bands at 284 and 1086 cm⁻¹, whereas dolomite shows peaks at 299 and 1097 cm⁻¹, and magnesite at 329 and 1094 cm⁻¹. The carbonate daughter minerals in the Sumdo inclusions consistently exhibit a peak combination at 283–284 and 1086 cm⁻¹ (Fig. S3), which is diagnostic of calcite and clearly distinct from other carbonate species. The aqueous fluid phase is identified by a broad Raman band between 3300 and 3600 cm⁻¹ (Fig. S3), corresponding to O–H stretching vibrations, indicating the presence of H$_2$O-rich fluid. Host omphacite is identified by its characteristic Raman peaks near 670 and 1012 cm⁻¹; in our analyses, peak positions at 672 and 1015 cm⁻¹ are consistently observed (Fig. S3), confirming the host mineral identity. Furthermore, no solid NaCl daughter minerals were detected in the fluid inclusions (which would be expected to show a strong Raman peak near 358 cm⁻¹). We therefore infer that NaCl is present as a dissolved component in the aqueous fluid phase. Furthermore, 3D maps were acquired via point-by-point scanning with spatial resolutions of 1 μm along the x-axis and 0.5 μm along the y- and z-axes. Acquisition times for individual points ranged from 0.5 to 2 s, with total scan durations of 5–20 h per fluid inclusion. This methodology has been successfully employed in our previous investigation of multiphase fluid inclusions from a UHP metamorphic vein in the Dabie orogenic belt, China[30].

### Reconstruction of fluid composition

Following the analytical protocol established in a previous study[30], we reconstructed the bulk compositions of the multiphase fluid inclusions investigated here. Volumetric reconstructions of multiphase fluid

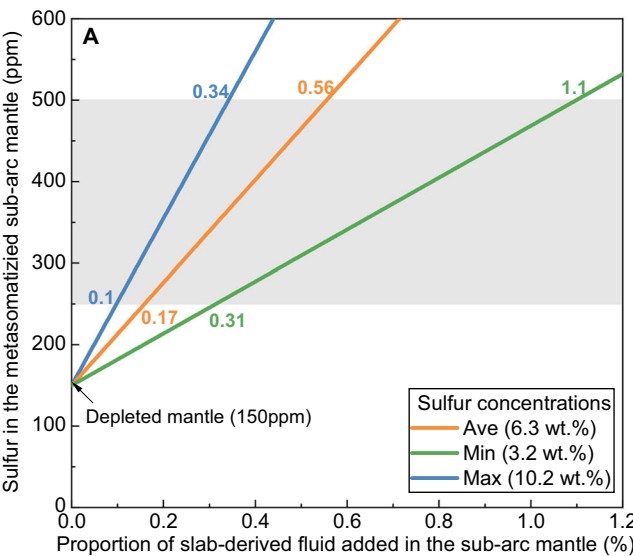

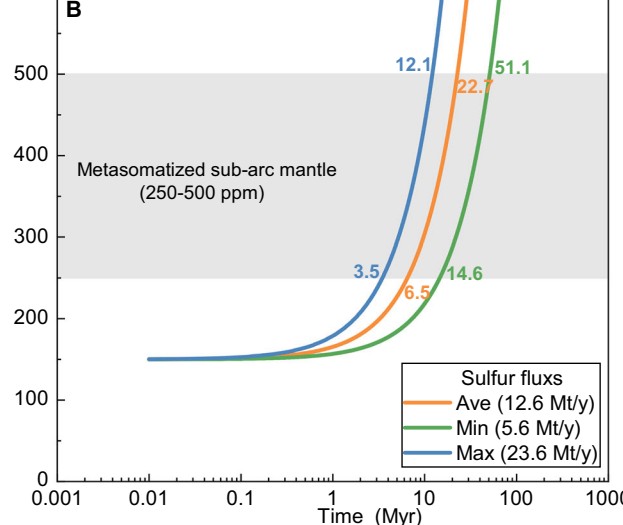

**Fig. 3 | Constraints on sulfur enrichment in the metasomatized sub-arc mantle. A** Modeled the relationship between the proportion of slab-derived fluid added to the sub-arc mantle and the resulting sulfur concentrations. Calculations using slab-derived fluids with sulfur concentrations of 6.3 wt% (average), 3.2 wt% (minimum), and 10.2 wt% (maximum), derived from fluid inclusion data. **B** Estimated timescales

required to increase sulfur concentrations in the sub-arc mantle from 150 ppm to target levels as a function of slab-derived sulfur flux. Calculations are based on 11.7 Mt/yr (average), 5.2 Mt/yr (minimum), and 21.9 Mt/yr (maximum) sulfur flux values. The grey band denotes the range of sulfur contents (250–500 ppm) observed in the metasomatized sub-arc mantle in both panels.

inclusions were carried out using WITec Project Plus and ImageJ software. Absolute volumes of individual daughter minerals were determined from the 3D maps (Supplementary Data 1) and combined with published mineral densities to calculate mass fractions for each phase. Idealized mineral chemistries were assumed for the reconstruction, including calcite ($CaCO_3$), chalcopyrite ($CuFeS_2$), and an aqueous brine composed of $H_2O$ with 15 wt% NaCl[37]. Details of the salinity measurements of brine have been reported previously[37], showing that omphacite-hosted multiphase fluid inclusions trapped at peak metamorphic conditions, corresponding to sub-arc depths, are systematically characterized by high salinities (10–22 wt% NaCl). For the purpose of bulk composition reconstruction, we therefore adopted an average salinity of 15 wt% NaCl. Densities of 2.71 g/cm³ for calcite, 4.10 g/cm³ for chalcopyrite, and 1.148 g/cm³ for the brine were used in the mass-balance calculations. This approach enables quantitative reconstruction of the bulk compositions of slab-derived fluids recorded by omphacite-hosted multiphase inclusions from the Sumdo eclogites. The robustness of the reconstructed fluid compositions is supported by several independent lines of evidence. In addition to the use of an established analytical framework[30]. First, analytical reproducibility was evaluated through independent repeat analyses of two representative fluid inclusions (Nos. 2 and 20). These repeat measurements yielded highly consistent daughter-mineral assemblages, phase volume proportions, and reconstructed bulk compositions (Supplementary Data 1 and 2), demonstrating good analytical reproducibility. Second, ternary compositional diagrams summarizing all 22 analyzed fluid inclusions show tight clustering of data points (Fig. S5), indicating limited compositional variability and strong internal consistency among inclusions. Finally, standard deviations are reported for all calculated major components, providing a quantitative estimate of the analytical uncertainties associated with the Raman-based reconstruction (Supplementary Data 2).

## Electron probe microanalysis of host omphacite across fluid inclusions

To evaluate potential post-entrapment interaction between the trapped fluid and the host mineral, a representative multiphase fluid

inclusion was selected for detailed analysis and carefully polished to near-surface exposure. The inclusion was initially ground using 10,000-grit silicon carbide sandpaper lubricated with ethanol until it approached the sample surface. Final polishing was performed on a polishing cloth using 10,000-grit carbon powder mixed with ethanol. Throughout the polishing process, the distance between the inclusion and the sample surface was continuously monitored under an optical microscope to avoid breaching the inclusion. Following sample preparation, the major element composition of the host omphacite surrounding the exposed fluid inclusion was analyzed by EPMA at the University of Science and Technology of China. Three linear analytical traverses were arranged to cross the inclusion boundary. Each traverse consisted of ten analytical points, with a spacing of 6 μm between adjacent points. The electron beam diameter was set to 3 μm. Analyses were performed at an accelerating voltage of 15 kV and a probe current of approximately 200 nA. The EPMA results are reported in Supplementary Data 3 and illustrated in Fig. S6. The data reveal minor but systematic variations in major-element compositions of the host omphacite surrounding the fluid inclusion. However, the magnitude of these variations is small, indicating that post-entrapment interaction between the host mineral and the inclusion was limited and did not significantly modify the bulk chemical composition of the trapped fluid.

## Thermodynamic modeling of the sulfur speciation in sub-arc fluids and dehydration of altered oceanic crust under Sumdo conditions

Thermodynamic calculations were carried out by Gibbs free energy minimization using Perple_X (version 7.1.13)[81]. Phase equilibria involving both solid and fluid phases were modeled in the Mn-Na-Ca-K-Fe-Mg-Al-Si-Ti-C-$O_2$-$H_2$-$S_2$ system. Thermodynamic data for condensed phase and molecular volatile species were taken from the revised Holland and Powell dataset (DS633, 2018)[82], and solute species data were adopted from the DEW model of Huang and Sverjensky[83](revised in 2019). Electrolytic fluid compositions were calculated using the lagged speciation algorithm, which ensures mass balance between coexisting solid and fluid phases. The thermodynamic properties of

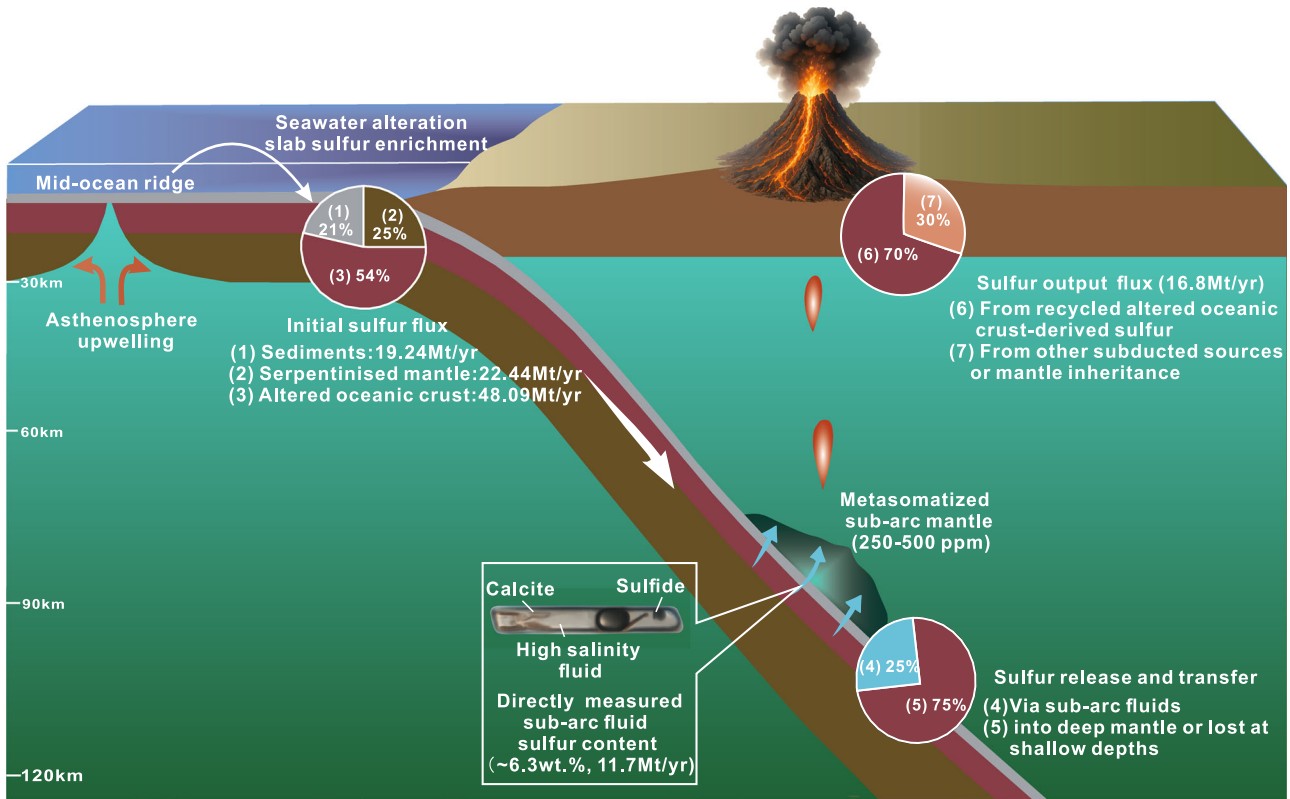

**Fig. 4 | Conceptual model of sulfur cycling in an oceanic subduction zone.**
Sulfur is initially enriched in altered oceanic crust (48.09 Mt/yr), sediments (19.24 Mt/yr), and serpentinized mantle (22.44 Mt/yr) through seawater alteration. Upon reaching sub-arc depths (-80–90 km), fluids released from the dehydrating altered oceanic crust—as recorded in omphacite-hosted fluid inclusions from the Sumdo eclogite (-6.3 ± 1.8 wt% sulfur)—mobilize sulfur into the overlying mantle

wedge. Approximately 25% of the slab-derived sulfur is released at these depths, enriching the metasomatized sub-arc mantle (250–500 ppm sulfur) over a time-scale of 6.5–22.7 Myr. This sulfur is ultimately returned to the surface via arc volcanism, accounting for -70% of the global sulfur output flux of -16.8 Mt/yr. The remaining sulfur may be retained in the slab and transported into the deeper mantle.

---

$H_2O$, $CO_2$, $CH_4$, $H_2S$, and $SO_2$ were described using the solvent reference state, whereas $CO$ and $C_2H_6$ were treated relative to the solute reference state owing to their negligible abundances. The equation of state of Pitzer and Sterner[84] was employed to calculate the thermodynamic properties of pure $H_2O$ and $CO_2$, while the modified Redlich–Kwong equation of state of Connolly and Cesare[85] was applied to all other solvent species and used to compute the activities of solvent components. The solid-solution models used in the calculations, together with their sources, are listed in Supplementary Data 4. Thermodynamic simulations were conducted using the bulk-rock composition of a representative Sumdo eclogite (T182[37,39]). The model input composition (in molar proportions) was specified as $Si = 20.3960$, $Al = 8.6834$, $Ti = 0.4259$, $Fe = 3.7864$, $Ca = 6.1017$, $Mg = 5.6123$, $K = 0.1146$, $Na = 1.7421$, $Mn = 0.0761$, $H_2 = 10.9999$, $C = 1.2661$, $S_2 = 0.0666$, and $O_2 = 42.8356$. The $Fe^{3+}/\Sigma Fe$ ratio (0.51), carbon content (2 wt%), and sulfur content (1500 ppm) were adopted from typical sulfur-rich altered oceanic crust compositions[60,86,87]. Following recent thermodynamic assessments[88,89], the $Mg(SiO_2)(HCO_3)^+$, $Fe(HCOO)^+$, and $H_2CO_3(aq)$ species were excluded to avoid potential artifacts from their unrealistically high predicted concentrations. Open-system fluid-fractionation modeling was applied along the P–T trajectory of progressive rock devolatilization[26]. The results of the thermodynamic modeling, including sulfur and carbon speciation and progressive dehydration behavior with increasing pressure and temperature, are summarized in Supplementary Data 4 and 5. At sub-arc depths of approximately 80–90 km, the modeled slab-derived fluids are dominated by oxidized sulfur species, primarily $SO_4^{2-}$ and $HSO_4^-$. In addition, the calculations indicate that approximately 60% of the water

initially stored in subducted altered oceanic crust is released at these depths, consistent with efficient slab dehydration under cold subduction conditions.

## Sulfur flux released from the altered oceanic crust at sub-arc depths

The peak metamorphic pressures recorded by omphacite in the Sumdo eclogite (-2.6–2.7 GPa[33]) correspond to sub-arc depths of approximately 80–90 km[36]. Our thermodynamic modeling constrained by the geological and P–T conditions of the Sumdo system indicates that -60% of the water initially stored in altered oceanic crust is released at these depths (Fig. S6). Treating the Sumdo system as representative of altered oceanic crust dehydration in cold subduction zones, we further integrate this dehydration fraction (-60%) with previously reported estimates of (i) the average $H_2O$ content in altered oceanic crust[61], (ii) a global subduction zone length of -55,000 km[62] and (iii) an altered crust thickness of 0.6 km[60] in order to quantify the global $H_2O$ flux released from altered oceanic crust at sub-arc depths (-80–90 km). This approach yields an $H_2O$ flux of -84.5 Mt/yr. Notably, this value is in good agreement with independent global estimates of $H_2O$ flux released from dehydrating altered oceanic crust (-91 Mt/yr[22]), supporting the robustness of our parameterization. The consistency of these values provides confidence in adopting this $H_2O$ flux as a key parameter for calculating sulfur flux released from altered oceanic crust at sub-arc depths (-80–90 km).

Using the sulfur and $H_2O$ concentrations reconstructed from the multiphase fluid inclusions in this study, together with the estimated $H_2O$ flux (-84.5 Mt/yr), we calculate the corresponding sulfur flux

released by altered oceanic crust-derived fluids at sub-arc depths as:

$$F_{\text{fluid}}^{\text{sulfur}} = \frac{F_{H_2O}}{C_{H_2O}} \times C_{\text{fluid}}^{\text{sulfur}} = 11.7 \pm 3.3 \, \text{Mt/yr} \qquad (1)$$

Here, $F_{\text{fluid}}^{\text{sulfur}}$ and $F_{H_2O}$ represent the sulfur flux and $H_2O$ flux, respectively. $C_{\text{fluid}}^{\text{sulfur}}$ and $C_{H_2O}$ denote the sulfur and water concentrations in slab-derived fluids. This calculation provides a direct and reasonable quantitative constraint on the sulfur flux released during dehydration of altered oceanic crust at sub-arc depths on a global scale.

### Desulfurization efficiency at sub-arc depths

To evaluate the extent of sulfur loss from the altered oceanic crust at sub-arc depths on a global scale, we compare the calculated sulfur flux in fluids ($F_{\text{fluid}}^{\text{sulfur}}$), with the total sulfur input from subducted altered oceanic crust ($F_{\text{input}}^{\text{sulfur}}$). The global sulfur input flux from the subducted altered oceanic crust is approximately 48.09 Mt/yr[4]. The desulfurization efficiency ($\eta$), defined as the proportion of sulfur released into slab-derived fluids at sub-arc depths, is given by:

$$\eta = \frac{F_{\text{fluid}}^{\text{sulfur}}}{F_{\text{input}}^{\text{sulfur}}} = 25 \pm 7\% \qquad (2)$$

This result implies that approximately one-third of the sulfur initially stored in the altered oceanic crust is mobilized into fluids at sub-arc depths, with the remainder either released at shallower levels or retained for recycling into the deeper mantle.

### Sulfur enrichment in the sub-arc mantle by infiltration of sulfur-rich fluids

To evaluate the potential of slab-derived fluids to enrich the sub-arc mantle in sulfur, we apply a mass-balance framework to estimate the fluid proportions required to elevate sulfur concentrations in metasomatized mantle domains to levels typical of arc-source regions (250–500 ppm[12,63]). Given that the mass of infiltrating fluids is much smaller than that of the mantle, the sulfur content of the metasomatized sub-arc mantle can be expressed as:

$$C_{\text{metasomatized sub-arc mantle}}^{\text{sulfur}} = C_{\text{depleted mantle}}^{\text{sulfur}} + C_{\text{fluid}}^{\text{sulfur}} \times f \qquad (3)$$

In this equation, the $C_{\text{metasomatized sub-arc mantle}}^{\text{sulfur}}$, $C_{\text{depleted mantle}}^{\text{sulfur}}$, and $C_{\text{fluid}}^{\text{sulfur}}$ represent the sulfur concentrations in metasomatized sub-arc mantle, depleted mantle, and sub-arc fluids, respectively. The variable $f$ represents the proportion of sulfur-rich fluids added to the metasomatized sub-arc mantle.

To further evaluate sulfur enrichment in the metasomatized sub-arc mantle over geological timescales, we estimate the mass of the sub-arc mantle domain between 70 and 100 km depth. Based on the total length of the global subduction zone (approximately 55,000 km[62]) and the estimated width of the sub-arc mantle per arc (approximately 150 km), the total volume of the 70–100 km sub-arc mantle is calculated to be $6.6 \times 10^{17}$ m³. With an average mantle density of 3300 kg/m³, the total mass of the 70–100 km sub-arc mantle is approximately $8.17 \times 10^{20}$ kg. Using the sulfur flux determined in this study, the time-dependent sulfur content of this reservoir can be calculated as:

$$C_{\text{metasomatized sub-arc mantle}}^{\text{sulfur}} = C_{\text{depleted mantle}}^{\text{sulfur}} + (F_{\text{fluid}}^{\text{sulfur}} \times t)/M_{\text{sub-arc mantle}} \qquad (4)$$

Here, $C_{\text{metasomatized sub-arc mantle}}^{\text{sulfur}}$ and $C_{\text{depleted mantle}}^{\text{sulfur}}$ are the sulfur concentrations in the metasomatized sub-arc mantle and depleted mantle. $F_{\text{fluid}}^{\text{sulfur}}$ is the calculated sulfur flux from sub-arc fluids, $t$ is the duration of fluid infiltration (typically taken as 1 Myr), and $M_{\text{sub-arc mantle}}$ is the mass of 70–100 km sub-arc mantle.

### Comparison of sulfur input via slab fluids and arc volcanic output

To evaluate the global significance of sulfur released from altered oceanic crust at sub-arc depths, we compared the calculated sulfur flux ($F_{\text{fluid}}^{\text{sulfur}}$) carried by the slab-derived fluids with the estimated global sulfur output from arc volcanic volcanism ($F_{\text{output}}^{\text{sulfur}}$), which is approximately 16.8Mt/yr[4]. The contributed proportion ($\Phi$) of slab-derived fluids' sulfur flux to the arc volcanic sulfur output can be calculated as follows

$$\Phi = \frac{F_{\text{fluid}}^{\text{sulfur}}}{F_{\text{output}}^{\text{sulfur}}} = 70 \pm 20\% \qquad (5)$$

This result indicates that slab-derived fluids may account for a dominant fraction of the sulfur released by arc volcanism. The calculated range overlaps well with previous estimates of ~40–70% derived from indirect constraints based on natural arc basalts[20], supporting the consistency and geological plausibility of our flux-based assessment.

## Data availability

All data supporting the findings of this study are available in the Supplementary materials and in a public repository (Figshare) at https://figshare.com/s/56febb3cbcea3e8f0e45.

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

## Acknowledgements

This work was supported by the National Natural Science Foundation of China (grant 42230304 to Y.X., grant 42302050 to D.-B.T. and grant 42302053 to Y.-B.L.), the National Key Research and Development Program of China (2023YFF0807103 to H.-Y.L.), and the Taishan Scholar Program of Shandong (tsqn202507278 to H.-Y.L.). We thank Dr. Wancai Li, Tingting Xiao, and Rui Shi for assistance with the EPMA analyses.

## Author contributions

Y.X., D.-B.T. conceived and designed the study. Y.-Y.W., H.-H.G. collected the samples. D.-B.T., H.-Y.L., D.-S.J., and X.-G.L. performed the geochemical analyses. Z.-L.G. conducted the thermodynamic modeling. D.-B.T. wrote the initial draft of the manuscript. Y.X., Y.-B.L., Carlos J. Garrido, and Timothy Kusky contributed to manuscript revision. Funding was acquired by Y.X., D.-B.T., Y.-B. L., and H.-Y. L. All authors discussed the results and contributed to the final manuscript.

## Competing interests

The authors declare no competing interests.

## Additional information

**Supplementary information** The online version contains Supplementary material available at https://doi.org/10.1038/s41467-026-71439-3.

