## [Transparent Peer Review File · Nature Communications]

Sulfur-enriched sub-arc fluids drive deep sulfur cycling in subduction zones

Corresponding Author: Professor Yilin Xiao

Version 0:

Reviewer comments:

Reviewer #1

(Remarks to the Author)

Dear Authors,

I read your paper with great interest. Your new omphacite-hosted fluid inclusion data from Sumdo eclogite belt are a useful addition to our understanding of fluid transfer and fluid compositions in subduction zone settings. I think this material is of broad interest. Your 3D Raman fluid inclusion data does provide direct evidence of sulfur-rich fluid compositions in a past subduction zone, which is great and confirms lot of model results already published in the literature.

However, I do have some reservations with the manuscript as it is, so I recommend that revisions are made to the paper so the results presented are more robust, also suggest making the paper a bit broader interest with the inclusion of other elements in parts of your model calculations.

One of my main criticism is the limited, and in my view, skewed, use of references in the paper. You often cite Li et al. 2020 (Nat Comms) as the reference for many of your statements and ignore a large number of papers that deal with sulfur in subduction settings and suggest (as you also do), that sulfur is strongly fluid-mobile and added to the mantle wedge from the slab in large quantities. I listed a few references in my detailed comments that you should probably include (some you already do, but don't always cite when you should). Your discussion should be more nuanced, as the way it is currently written, it feels like you are the first to prove efficient S transfer via slab-derived fluids, but there is already plenty of evidence for this from arc volcanoes and model results. Limitations of your data are not acknowledged – surely, one sample from one eclogite belt will be representative of one P-T-X subduction path, not global variability.

Secondly, I'm not convinced that your inclusions are primary (they can be secondary based on the evidence you provide), and this needs to be addressed or acknowledged in the text. I'm also not convinced that your inclusions have not undergone post-entrapment modification. You kind of brush this under the carpet in the text and provide no evidence that rules out Ca scavenging from the host (to form calcite from a CO₂-rich fluid), or FI and host electron exchange (between Fe³⁺ present in omphacite and S⁶⁺ that may have been in the fluid). In summary, I'm not sure that your inclusions were reduced, as you suggest. This needs to be considered and addressed, as oxidised fluids thought to carry large quantities of S, based on most recent DEW models, and may fit better with your data.

Your arguments that Cl influences S behaviour at the end of the manuscript feel very speculative. These are two fluid mobile elements, and Cl is elevated in all subduction zone magmas, while S is not always enriched. So I don't really feel there is any causality here, and that you provide sufficient proof that there is.

I'm not sure who your flux models work – is this for a fictional subduction zone, or for a global S flux from the slab? It needs to be better explained.

Data tables are provided as a Word document, which is not particularly useful for other researchers who may want to use your data – you probably have an excel/csv table for these – please provide data in that format instead.

Here are my detailed comments, line-by-line:

Lines 47-48: While the 20-30% fluid contribution from the slab to the mantle wedge, and subsequently to the arc, is indeed the value in Li et al. (2020) you cite, but this is one of only few papers that argue for low amounts of S being transferred from

the slab to the wedge, and there are many others that say the opposite. For example, in Central America, we estimated 40-70% sulfur addition to the mantle wedge from the slab in Taracsák et al. 2023. The way it is currently written implies there is only this 20-30% range in the literature, which is not the case.

Line 51: This implies that people argued for extra S in the mantle before subduction. I have not seen papers suggesting high S contents in the subarc mantle (relative to normal, depleted upper mantle) under arcs prior to subduction-derived S addition. Is there a reference for this?

Line57: a minor issue but melt inclusion paper you cite here (Muth & Wallace 2021) used tephra samples, not lavas.

Line70-72: you cite Evans 2012 (a redox focussed review paper) and Bekaert et al. 2021 (a volatile cycle review paper), rather than papers that look at subducting S budgets (de Moor et al. 2022 Chem Geol, Peccia et al. 2025 Chem Geol), that are much more detailed and more relevant to your work. Crucially, both of these papers suggest it is the primary igneous crust that has the largest S budget followed by the AOC – and the total igneous package (AOC+ primary) are the most important sources of S. Is there any proof that your eclogites represent AOC, or can they be metamorphosed primary crust (unaltered)?

Line 99: the presence of pyrite is interesting. Looking at the recent model results of Beaudry & Sverjensky 2024 (G-cubed), who extended the Deep Earth Water model's calibration range and included oxidised S species, they suggest that under conditions similar to what you imply for these samples (~3GPa, 600-800 Celsius), pyrite is present at FMQ-1 to FMQ+2 (at higher T, it is more restricted to FMQ+0.5 to +1.5). Critically, their model results provide S contents in fluids in equilibrium with mafic eclogite that reach the S contents you propose based on your Fis (and even higher values), so it is a very relevant paper to your results, and you currently don't cite or integrate this into your paper. I think you need to carefully consider their findings regarding the S carrying capacity of fluids at higher fO₂ than those modelled in Li et al. 2020, as they predict up to an order of magnitude more S in the fluid.

Line125-128: the reconstructed fluid compositions are very interesting, particularly with respect to their high Cu contents. This is because, as far as I'm aware, Cu has been suggested not to mobilise during slab devolatilisation (e.g. Jenner, 2017 Nature Geo). Your paper could be more broad interest I think if you explored other elements alongside S, such as Cu, H₂O or Cl. For example, to see if your FIs have enough H₂O to enrich the mantle wedge in H₂O and match estimates for mantle wedge H₂O contents that range from 0.3 to 1wt% in the Marianas (Kelley et al. 2010 J. Petrol). The 0.1% to 1% fluid addition in Fig 3A seems a bit too low for that. Also, how much Cu would be added to the mantle under the same circumstances, considering your Fis have 5 wt%, and the upper mantle only ~30 ppm?

Line 136: I have some issue with your interpretation of the FIs being primary. In line 110, you mention FIs are parallel with the C axis (a cleavage direction for cpx). You also mention they are confined to cores and have negative crystal morphologies. But together, these observations cannot rule out that the inclusions are in fact secondary, entrapped in cracks within the omphacite after formation, followed by further crystal growth along the rims. Negative crystal morphologies alone cannot be used to argue for a primary origin (see Goldstein 2003 review on FI petrography). In any case, you need to provide some further arguments why your inclusions are representative for peak metamorphic conditions, considering a potential secondary origin relative to omphacite cores.

Line 140: I doubt that you can rule out post entrapment re-equilibration. H⁺ diffuses rapidly within cpx, so you could gain or lose it depending on H activity around the grains. Fe²⁺ can also diffuse in and out. Both of these elements could influence the redox within your inclusions. This leads me to think that the S may have been oxidised in the fluid originally, not reduced, and the chalcopyrite formed due to post-entrapment processes. Can you rule this out, and if yes, how? If not, this needs to be considered.

Line145-146: I don't understand this sentence. What is replaced? Feels like the subject of the sentence is not defined.

Line157: you say that your S contents at 3-10 wt% is similar to the literature range, but in fact it is narrower by an order of magnitude, and you exclude the lower end. So rather than saying it is similar, I would say it is more restricted.

Line160-161: Well, you argue that your results are more faithful than thermodynamic models to the real fluid composition. But surely, your one sample is restricted to one P-T-X path, so hardly a representation of global subduction zone variability that include hot and cold, steep and shallow, and old and young slabs (see e.g. Syracuse et al. 2010). The restricted S content range restricted range may be representative of the fact that you only look at a single P-T-X.

Lines164-167: While I very much agree with this statement, you kind of put this here in a vacuum, and somewhat feels like you are implying this is a new idea (and cite no other paper that suggested this). In fact, lots of papers argued for extensive S transfer from slab to mantle wedge, both old and new (Métrich et al. 1999 EPSL, de Hoog et al. 2001 EPSL, Walters et al. 2020 GPL, Chowdhury & Dasgupta 2019 Chem Geol, Beaudry & Sverjensky 2024 G-cubed, Muth and Wallace 2022 EPSL, Taracsák et al. 2023 EPLS, Kawaguchi et al. 2022 JPetrol, just to list a few). Yes, your results confirm this, and are valuable, but they add to existing evidence that you should mention here.

Line169: I think this subsection title misses a word, or mechanism is not the best way to put it. "Formation of ..." may be better?

Line173: reference 33 is in fact a review of a paper in Nature Geo, not the paper itself that presents the results – that would

be Farsang and Zajacz 2024 Nature Geo.

Line176: again, oxidised species are not considered here, yet these can carry more S than reduced species – see earlier comments.

Lines178-180: I already mentioned this earlier; but pyrite is not an indicator of low fO_2 (pyrrhotite would be), and in fact a higher fO_2 may be needed for pyrite. Can you be sure that the cpy is not formed by post entrapment reduction of S, considering Fe and S can readily exchange electrons? Omphacite can contain aegirine and Fe^{3+} - in fact, Weller et al. 2016 measured a $Fe^{3+}/Fe(T)$ of 0.33-0.45 in peak omphacite, so there is plenty of it to reduce the little S (with respect to total mass) present in the FIs. And a similar argument can be made for calcite – can the Ca scavenged from the omphacite (it has 18 wt% CaO)? A similar process is described for opx-hosted, CO_2 -rich FIs from mantle xenoliths (Berkesi et al. 2012 EPSL), where reactions between the fluid and the host formed carbonates.

Line228-230: if I understand correctly, you argue that sulfur is more mobile because of the high Cl content of fluids, and that there is a coupling effect. Yet, there is plenty of evidence that S is fairly fluid mobile as it is, especially under oxidising conditions. Cl is always very fluid mobile (elevated in all arcs) – so this can simply be the case of two fluid mobile elements behaving similarly, because they have similar affinity to fluids under certain conditions? The citation you provide in line 227 does not add to your argument – it is a melt inclusions paper from 22 years ago, that highlights there is a lot for Cl and S in arc melt inclusions, and they modelled a fluid composition based on mass balance – it is not an observed value for fluid S and Cl content.

Line240-242: Even for one arc, these estimates vary a lot – see de Moor et al 2022 for central America, who estimate that between 10-60% of S is lost to the mantle wedge from the slab. So, most S may enter the deep mantle, but also, most may go into the arc.

Lines248-271: the flux calculations are confusing because you do not actually explain what you want to show, and how you do it; are these values global fluxes, or just for a specific subduction zone? Is this for a certain thickness of crust. It feels a bit confusing.

Line275: the paper you cite on sediment melt S solubility says 1800 ppm at the SCAS – which is not that high (similar to a basalt at the SCSS) – combine this with the fact that sediments rich in S are usually reduced (e.g. Peccia et al. 2025 Chem GEol), makes me question if sediments melts can in fact carry substantial amounts of S as you state.

Line318: I cannot find this water flux anywhere in the referenced paper. You should really explain what this “flux” is, not just say that this is the water flux at 90 km depth (which means not much – every subduction zone will have a different H_2O flux due to different P-T-X and crustal thickness). Is this for a certain thickness of crust dehydrating?

Line329: Same question applies for the sulfur input flux – what does this value represents?

Line364-365: This value does not significantly exceed what was suggested for Central America (between 40-70% of S being slab-derived in arcs) in Taracsák et al. 2023 – so in a sense your results confirm what was already suggested based on arc melt inclusion data.

Methods section: This is very little detail considering these Raman analyses are the base of the paper - what peaks/spectral features were used to identify phases? How sure can we be that each phase is a pure mineral phase and not something similar, yet compositionally different? (e.g. cpy vs other sulfides, calcite vs dolomite or magnesite?). While I appreciate that you provide four representative spectra in the supplementary, I think you should give a more detailed summary here.

Line 535: “Approximately this sulfur is eventually” sounds strange, I suggest rewording this sentence.

Figure 2: It would be interesting to add a ternary plot with water, cpy, and cc as the three endmembers – if the inclusions have a common origin, the fractions of the three phases should be similar across all inclusions. Maybe this can go in as a supplementary figure.

Table 1: you could add standard deviation and minimum/maximum values, as you use those in your Fig. 3.

I hope these comments are helpful and will assist improving your ms.

Best wishes,
Zoltán Taracsák

Reviewer #2

(Remarks to the Author)

This manuscript presents the first in situ quantification of sulfur contents in natural slab-derived fluids from sub-arc depths, based on multiphase fluid inclusions hosted in omphacite from UHP eclogites in the Sumdo belt, southern Tibet. The authors convincingly demonstrate that these fluids contain exceptionally high sulfur concentrations (~6 wt.%), potentially accounting for up to 75% of the global arc volcanic sulfur output. This is an important contribution to understanding sulfur cycling agents and fluxes in subduction zones, and the study is timely, original, and methodologically sound. The dataset is

of high quality, the interpretations are generally well supported, and the implications are broadly relevant to geochemistry, subduction dynamics, and volatile transfer processes. Overall, the manuscript is suitable for publication in Nature Communications after moderate revisions in order to improving methodological transparency and quantitative rigor.

Major Comments

1. Uncertainty and reproducibility of Raman-based quantification

The 3D Raman mapping and volumetric reconstruction form the technical foundation of this study. However, no explicit uncertainty propagation is provided for the calculated bulk compositions (Table 1). Please quantify and discuss the analytical uncertainties and reproducibility of the Raman-based measurements.

2. Variability among subduction settings

Please discuss potential variability of slab-derived sulfur contents among different subduction zones (e.g., Alps, Caledonides, Japanese arcs), and clarify the limitations of extrapolating the Sumdo data to global sulfur flux estimates.

3. Sulfide mineralogy and sulfur speciation

Chalcopyrite is the only sulfide phase observed in the fluid inclusions, whereas pyrite dominates in the host eclogite. Why is this the case? In addition, no oxidized S-bearing volatiles (e.g., SO_4^{2-}) were detected in the omphacite inclusions, although experimental and thermodynamic models suggest that oxidized sulfur species should be more mobile under subduction conditions. Please discuss this discrepancy.

4. Role of calcite and carbonate components

What is the role of calcite in the fluid inclusions? Could CO_3^{2-} enhance sulfide solubility instead of (or in addition to) NaCl? This should be addressed quantitatively or at least conceptually.

5. Potential overestimation of global sulfur fluxes

The limited omphacite inclusions from UHP eclogites in the Sumdo belt may represent locally sulfur-enriched portions of the subducting slab. Consequently, using these inclusions to estimate global slab-derived sulfur contributions might lead to overestimation. Please discuss this limitation explicitly.

Minor Comments

- Line 118: Please include full microscopic images of the mineral assemblage in the Supplementary Materials and specify the trapping stage of the fluids (during subduction or exhumation). Are the omphacites vein minerals, as reported in other subduction settings?
- Lines 144–145: Clarify why chalcopyrite appears as the only sulfide phase within the inclusions, while pyrite dominates the host eclogite. Does chalcopyrite have higher solubility under subduction conditions?
- Line 146: The pronoun “that” is ambiguous—does it refer to pyrite or chalcopyrite? Please clarify whether decomposition is pressure- or temperature-driven, and cite any reports of pyrite in metamorphic fluid inclusions.
- Line 176: Explain why reduced sulfur species (S^{2-}) are released rather than oxidized ones (SO_3^{2-}). Note that reduced sulfur may be re-trapped by Fe^{2+} in the host rocks.
- Lines 221–222, 235–236: Provide details about the salinity measurements and indicate whether NaCl is present in the Sumdo inclusions. Could CO_3^{2-} enhance sulfide solubility?
- Line 234: The discussion of “supercritical-like” fluids is speculative without supporting evidence. Use this term cautiously or replace it with “near-critical, solute-rich fluids,” and cite relevant experimental studies (e.g., Mibe et al., 2011).
- Lines 248–250: When calculating the slab-derived sulfur flux, it would be helpful to provide the estimated water flux for the Sumdo subduction zone, rather than relying solely on global averages.
- Line 295: Include details of laser power and acquisition parameters in the Methods section to ensure reproducibility.
- Lines 328–329: The global sulfur input flux from subducted altered oceanic crust (~48.09 Mt/yr) may not directly apply to the Sumdo subduction zone. Please provide uncertainty ranges (e.g., $75 \pm ?$ %) to reflect the robustness of your global estimate.

Version 1:

Reviewer comments:

Reviewer #1

(Remarks to the Author)

Dear Authors,

I read the revised version of the ms. You sufficiently addressed the major comments, and I think the manuscript in its present form is much more convincing and suitable for publication in Nature Communications. I appreciate the extra modelling work and EPMA analyses you carried out. I have some minor comments and suggestions to the text where I think you should be more nuanced as I indicate below (these should be simple to do by adding a sentence or two)—once these are sorted, I recommend publication.

Line62: Maybe reword the start of the sentence starting with “more fundamentally” – I suggest “It is also possible such discrepancy arises”...

Line125-142: It is good that you added this segment - but I still have an issue around the use of “primary origin” - which strictly means that the FIs were entrapped during growth of the omphacite cores. Crosscutting inclusions with homogeneous fill may still be secondary relative to the core, placing FI formation between growth of omphacite cores and rims. I accept the arguments here that these FIs represent near-peak P-T conditions (particularly as these are in the core only), and

representative of fluids lost from slabs - but you can say that without confirming primary origin (as you do in lines 139-142), or by mentioning that even if the FIs formed after the omphacite cores crystallised, they predate the rims, which is firm evidence of eclogite facies origin.

Line164: again, use of primary fluid inclusion. I would leave primary out, and just state that these fluids were present concurrent to omphacite crystallisation at high P-T.

Line181-184: I agree that post-entrapment addition/loss of sulfur and Cu can be ruled out - but for elements like Ca and Fe, the contribution from the host may still be non-negligible. I appreciate that you carried out the EPMA work, but an issue that remains is that the silicate-host may have undergone diffusive re-equilibration too (partially or fully). I recommend changing the text here slightly to add nuance (i.e. add that modification was negligible for element other than Ca and Fe).

Line198-202 I don't think these two sentences add to your argument, considering that the quoted experiment/model based S range is, while broader, it still fully overlaps with your FIs. I would delete these two sentences and only leave in the revised text.

Line276: delete squarely

Line278: mention where Dabie Shan is for broader context, and what subduction system it represent.

Line290: Maybe this is one of my last real concern with the ms - I think it would be important to highlight here that Ca contents may be affected by exchange with the host here, and also the fact that you did not measure Na in the brine, but assumed a 15% salinity based on a published dataset that had a range of 12-22%. In line 291 you state these concentrations are observed, but really Na is just estimated or implied. You should point out these limitations at the end of the paragraph.

Line311-312: delete "capable of transporting high solute loads (solute-rich already implies this)

Line371-373: How does the difference between isotope-based models of fluid fraction (5%) reinforces the robustness of your estimate of 0.2-0.6%? To me this looks like a 10x discrepancy. Maybe delete "thereby reinforcing the robustness of our mass-balance constraints"

Line545: You state that you calculated " the global H₂O flux from sub-arc depth", but to me it seems you just quantify H₂O from the AOC, based on your previous sentence – isn't this a global flux estimate for the AOC only, not including sediments or serpentinites?

Best wishes,
Zoltán Taracsák

Reviewer #2

(Remarks to the Author)
Dear Editor,

I just have checked this revised manuscript carefully. All my questions and suggestions have been accepted and changed in the revised version. I suggest to accepted for the publication in NC.

Sinesely,

Lifei Zhang

Point-by-point response

Reviewer 1 Dr. Zoltán Taracsák's comment

I read your paper with great interest. Your new omphacite-hosted fluid inclusion data from Sumdo eclogite belt are a useful addition to our understanding of fluid transfer and fluid compositions in subduction zone settings. I think this material is of broad interest. Your 3D Raman fluid inclusion data does provide direct evidence of sulfur-rich fluid compositions in a past subduction zone, which is great and confirms lot of model results already published in the literature. However, I do have some reservations with the manuscript as it is, so I recommend that revisions are made to the paper so the results presented are more robust, also suggest making the paper a bit broader interest with the inclusion of other elements in parts of your model calculations.

Reply: We sincerely thank Dr. Zoltán Taracsák for the careful, constructive, and positive review of our manuscript. We greatly appreciate the reviewer's constructive comments and thoughtful suggestions and have thoroughly revised the manuscript accordingly. All changes are clearly indicated in the revised version (shown in blue), and detailed point-by-point responses are provided below. For the reviewer's convenience, we summarize the major revisions here.

(1) We have substantially expanded the citation of literature related to sulfur cycling in subduction zones in the revised manuscript and have incorporated these references at appropriate locations throughout the text.

(2) To assess the potential effects of post-entrapment modification, we selected a representative fluid inclusion, polished it to the surface, and conducted systematic electron probe microanalysis (EPMA) traverses across the surrounding host omphacite. By quantifying major-element variations in the host mineral as a function of distance from the inclusion, we find that although limited post-entrapment modification may have occurred, its overall magnitude is minor. Importantly, sulfur is absent in the host omphacite, implying that any post-entrapment modification would, if anything, lead to a minor underestimation of the original sulfur contents of the trapped fluids. Therefore, this process does not compromise, but rather reinforces, our conclusion that sub-arc fluids were sulfur-enriched.

(3) Following the reviewer's suggestion, we have clarified the sulfur speciation of the initially entrapped sub-arc fluids and explicitly discussed the potential influence of fluid–host interaction. (i) EPMA profiles reveal slightly lower Fe contents in omphacite adjacent to inclusions relative to more distal regions (0.4 wt.% FeO variations), suggesting limited Fe exchange between host and inclusion. Such redistribution could have locally modified sulfur speciation within the inclusions, potentially shifting it from initially oxidized forms toward reduced species during or after entrapment. (ii) To further constrain the primary sulfur speciation, we performed thermodynamic modeling using the DEW framework under the geological P–T conditions of the Sumdo system. These calculations indicate that slab-derived fluids at sub-arc depths are dominated by oxidized sulfur species (e.g., SO_4^{2-} , HSO_4^-) at the time of entrapment. Importantly, although sulfur speciation may be locally modified by fluid–host interaction, the total sulfur contents of the fluids remain unaffected. Accordingly, our central conclusion—that slab-derived fluids at sub-arc depths can contain exceptionally high sulfur concentrations—remains robust.

(4) We have strengthened and clarified the discussion regarding extrapolation from the Sumdo system to broader subduction contexts by explicitly stating the assumptions, rationale, and

limitations. In summary: (i) The Sumdo eclogites record a cold subduction regime, which represents the dominant style of subduction operating globally. As such, the sulfur systematics documented here are expected to be broadly relevant to other cold subduction zones. (ii) The studied inclusions capture fluid compositions at sub-arc depths, a crucial window for slab-derived sulfur release. We further clarify that sulfur-rich sub-arc fluids are not unique to the Sumdo system, but have been reported in multiple subduction settings worldwide. Moreover, our measurements not only validate previous experimental and theoretical predictions of elevated sulfur contents in sub-arc fluids but also provide more restricted natural constraints on the range of sulfur concentrations in slab-derived fluids. (iii) We performed new thermodynamic modelling constrained by the geological context and P–T conditions of the Sumdo system to quantify slab dehydration. The results indicate that ~60% of the water initially stored in altered oceanic crust is released at sub-arc depths. Treating the Sumdo system as representative of altered oceanic crust dehydration in cold subduction zones, and integrating this dehydration fraction (~60%) with the average H₂O content of altered oceanic crust, global subduction zone length, and crustal thickness, we estimate a global H₂O release flux of ~84.5 Mt/yr at ~80–90 km depth. This estimate is consistent with independent literature constraints (~91 Mt/yr), supporting the robustness of our water-flux parameterization. The independently constrained H₂O flux thus provides a critical parameter for refining estimates of sulfur flux transported by slab-derived fluids at sub-arc depths. (iv) Based on the calculated sulfur fluxes, we further quantify the efficiency of slab-derived sulfur release at sub-arc depths and evaluate its contribution to sulfur enrichment in arc systems. These estimates are in quantitative agreement with independent experimental, theoretical, and geochemical constraints, supporting both the reasonableness of our parameterization and the reliability of extrapolating results from the Sumdo system to global cold subduction zones.

(5) In response to the reviewer's insightful comment regarding the elevated copper concentrations measured in the fluid inclusions, we have newly incorporated a concise discussion of their broader geochemical implications. Our results highlight the role of sulfur-rich oxidized sub-arc fluids in promoting efficient copper mobilization, with significant implications for the formation of porphyry copper deposits in arc magmatic systems.

In addition to the major revisions outlined above, we have carefully addressed all other constructive comments and suggestions in the revised manuscript. We sincerely hope that the revised version meets the reviewer's expectations. We are grateful to Dr. Zoltán Taracsák for the thoughtful and insightful comments, which have substantially strengthened the clarity, rigor, and overall quality of this work.

General comments

(1) One of my main criticism is the limited, and in my view, skewed, use of references in the paper. You often cite Li et al. 2020 (Nat Comms) as the reference for many of your statements and ignore a large number of papers that deal with sulfur in subduction settings and suggest (as you also do), that sulfur is strongly fluid-mobile and added to the mantle wedge from the slab in large quantities. I listed a few references in my detailed comments that you should probably include (some you already do, but don't always cite when you should). Your discussion should be more nuanced, as the way it is currently written, it feels like you are the first to prove efficient S transfer via slab-derived fluids, but there is already plenty of evidence for this form arc

volcanoes and model results. Limitations of your data are not acknowledged – surely, one sample from one eclogite belt will be representative of one P-T-X subduction path, not global variability.

Reply: We greatly appreciate the reviewer's constructive and thoughtful comments. We fully agree that these aspects required clearer treatment, and we have revised the manuscript accordingly.

(1) We acknowledge that the original version of the manuscript placed disproportionate emphasis on Li et al. (2020). In the revised manuscript, we have substantially expanded and rebalanced the citation of the literature on sulfur cycling in subduction zones. This includes the references suggested by the reviewer, as well as additional key studies published up to 2026.

(2) We have revised the manuscript to avoid implying that our study is the first to demonstrate efficient sulfur transfer from the slab to the mantle wedge. Instead, we now clearly emphasize that our primary contribution is the first direct, in situ quantification of sulfur concentrations in natural slab-derived fluids at sub-arc depths, achieved through three-dimensional (3D) Raman characterization of multiphase fluid inclusions. Importantly, our findings provide a critical observational bridge linking experimental constraints, thermodynamic models, and sulfur-enriched signatures observed in arc magmas, thereby strengthening and integrating existing frameworks rather than replacing or contradicting them.

(3) Finally, we now explicitly acknowledge the limitations associated with extrapolating results from the Sumdo system to global subduction zones. We recognize that subduction zones worldwide exhibit substantial variability in thermal regimes (e.g., cold versus hot subduction), slab dip angles, and convergence rates, all of which may influence the behavior of sulfur during subduction. These limitations are now clearly stated in the revised manuscript.

At the same time, we also clarify why the Sumdo system nevertheless provides meaningful constraints on the deep sulfur cycle at the global scale, based on the following considerations.

(i) The Sumdo system records a cold subduction regime, which represents a widespread and geodynamically important class of subduction zones operating globally. Although results from a single system cannot capture the full range of global variability, cold subduction zones contribute substantially to global volatile recycling, allowing the extrapolating results from the Sumdo system provide broadly applicable constraints on subduction zone sulfur budgets. (ii) Sub-arc depths constitute a primary window for sulfur release from the subducted slab, and the samples investigated here directly record this critical stage of sulfur transfer. Notably, the sulfur concentrations quantified in our study fall within the range predicted by previous experimental and theoretical studies, supporting the robustness and reliability of our observations. (iii) Using geological constraints from the Sumdo system, we performed thermodynamic calculations to estimate slab dehydration extents and corresponding H₂O fluxes at sub-arc depths (80–90 km). The resulting H₂O fluxes are consistent with independent estimates reported in the literature, further supporting the representativeness of the Sumdo system within the context of cold subduction. (iv) By integrating the measured sulfur concentrations with the calculated H₂O fluxes, we estimated sulfur fluxes carried by sub-arc fluids and evaluated the efficiency of slab sulfur release at sub-arc depths and its contribution to sulfur enrichment in arc systems. These estimates are in quantitative agreement with independent experimental, theoretical, and geochemical constraints, supporting a cautious yet reasonable extrapolation of our findings from the Sumdo system to global subduction zones.

(2) Secondly, I'm not convinced that your inclusions are primary (they can be secondary based on the evidence you provide), and this needs to be addressed or acknowledged in the text. I'm also not convinced that your inclusions have not undergone post-entrapment modification. You kind of brush this under the carpet in the text and provide no evidence that rules out Ca scavenging from the host (to form calcite from a CO₂-rich fluid), or FI and host electron exchange (between Fe³⁺ present in omphacite and S⁶⁺ that may have been in the fluid). In summary, I'm not sure that your inclusions were reduced, as you suggest. This needs to be considered and addressed, as oxidised fluids thought to carry large quantities of S, based on most recent DEW models, and may fit better with your data.

Reply: We thank the reviewer for these insightful and important comments. We agree that the origin of the fluid inclusions, the possibility of post-entrapment modification, and the sulfur speciation require careful evaluation and clearer treatment. In the revised manuscript, we have substantially revised the text to explicitly address these issues, as summarized below.

(1) We have clarified and strengthened the criteria supporting a primary origin for the studied multiphase fluid inclusions. The revised manuscript (Lines 125–142) now states: “Multiple independent lines of evidence support a primary origin for the omphacite-hosted fluid inclusions. First, the fluid inclusions are confined to the cores of omphacite grains and do not form trails that crosscut the host crystals (Fig. 1), as would be expected for secondary inclusions trapped along healed fractures. Detailed petrographic observations further reveal no microfractures or healed cracks in the host omphacite that could have acted as pathways for late-stage fluid infiltration (Fig. S2). Second, the fluid inclusions commonly occur as isolated or as arrays parallel to the crystallographic c-axis of omphacite (Fig. 1). Individual fluid inclusions are typically ~5–60 μm in length and ~2–8 μm in width and display a consistent columnar morphology that mimics the elongated crystallographic habit of the host mineral. This negative-crystal-like geometry is consistent with primary entrapment and provides supporting, though not diagnostic, evidence for a primary origin³⁸. Third, the fluid inclusions show uniform filling degrees of ~80–90% at room temperature and contain consistent proportions of liquid, solid, and vapor phases (Fig. 1), a feature that is inconsistent with heterogeneous secondary trapping. Taken together, these textural and morphological characteristics indicate that the studied inclusions represent primary fluid inclusions. Combined with the fact that omphacite crystallized exclusively at the peak stage of UHP conditions³⁷, corresponding to sub-arc depths³⁶, these observations suggest that omphacite-hosted fluid inclusions provide the most reliable recorders of slab-derived fluid compositions at sub-arc conditions.”

(2) We fully agree that potential post-entrapment modification must be carefully assessed. To directly address this concern, we selected a representative multiphase fluid inclusion, polished it to the surface, and conducted systematic EPMA transects across the host mineral (omphacite) surrounding the fluid inclusions. By quantifying major-element variations as a function of distance from the inclusion, we find that some degree of post-entrapment modification did occur, but its overall extent is limited (Fig. S6). Importantly, sulfur is absent in the host omphacite. Therefore, any fluid–host interaction during post-entrapment modification would not result in a decrease of the sulfur concentrations preserved within the inclusions. Consequently, our central conclusion—that the trapped fluids were sulfur-rich—remains robust and unaffected by post-entrapment processes.

(3) Furthermore, we now explicitly acknowledge that post-entrapment processes may have influenced sulfur speciation within the inclusions. The newly added EPMA data reveal slight variations in Fe contents in omphacite adjacent to the investigated fluid inclusions, which may reflect a weak Fe exchange between the host mineral and the inclusions (Fig. S6). Such electron exchange could plausibly modify sulfur speciation after entrapment. To address this more rigorously, we have newly conducted a thermodynamic modeling using the DEW framework, constrained by the geological and P–T conditions of the Sumdo system. These calculations indicate that sulfur in slab-derived fluids at sub-arc depths is dominantly present as oxidized species (e.g., SO_4^{2-} , HSO_4^-) (Fig. S7). Accordingly, we have revised the manuscript to clarify that the primary slab-derived fluids at sub-arc depths were likely oxidized and sulfur-rich, which may be reduced in the fluid inclusions after entrapment.

We sincerely thank the reviewer once again for raising these important and constructive points. Addressing these issues has substantially improved the clarity, rigor, and overall robustness of the manuscript!

Fig. S6 in the supplementary information. Major-element variations in host omphacite across a representative multiphase fluid inclusion. (A-G) Electron probe microanalysis transects showing variations in FeO, CaO, SiO₂, MgO, Al₂O₃, Na₂O, and TiO₂ (wt.%) in host omphacite across a representative multiphase fluid inclusion. Three parallel analytical transects (Lines 1–

3) were conducted across the inclusion, with the grey shaded region marking the position of the fluid inclusion. (H) Schematic illustration showing the relative positions of the three transects and the fluid inclusion polished to (near-)surface. Overall, major-element compositions of omphacite show only slight variations across the inclusion (e.g., ~0.4 wt.% for FeO and ~0.6 wt.% for CaO), indicating that post-entrapment modification of the host mineral is minor.

Fig. S7 in the supplementary information. Thermodynamic modeling of sulfur speciation in slab-derived fluids constrained by the Sumdo system. The results show that slab-derived fluids at sub-arc depths are dominated by oxidized sulfur species (e.g., SO_4^{2-} , HSO_4^-).

(3) Your arguments that Cl influences S behaviour at the end of the manuscript feel very speculative. These are two fluid mobile elements, and Cl is elevated in all subduction zone magmas, while S is not always enriched. So I don't really feel there is any causality here, and that you provide sufficient proof that there is.

Reply: We are grateful to the reviewer for raising this important point. We agree that the original discussion placed too much emphasis on a potential causal role of Cl in controlling sulfur behavior. In the revised manuscript, we have substantially restructured this section.

(1) Following the reviewer's suggestion, and guided in particular by the recent experimental and thermodynamic study of Beaudry and Sverjensky (2024), we now interpret the trapped fluids as being dominated by oxidized sulfur species and focus instead on the role of major cations (Ca and Na) in dissolving sulfur in fluids at sub-arc conditions. Beaudry and Sverjensky (2024) demonstrate that oxidized sulfur can be efficiently mobilized in high-pressure fluids through complexing with Ca and Na (e.g., CaHSO_4^+ and Na_2SO_4^0), resulting in significantly enhanced sulfur solubility under sub-arc conditions. Importantly, this revised interpretation is fully consistent with our observations of elevated CaO (10.16 wt.%) and Na (3.22 wt.%), and

high sulfur contents in the multiphase fluid inclusions. We therefore believe that the revised discussion presents a more appropriate interpretation of the sulfur enrichment mechanism in slab-derived fluids.

(2) As noted by the reviewer, chlorine is itself highly fluid-mobile, and the apparent covariation of sulfur and chlorine in arc systems may simply reflect their shared affinity for slab-derived fluids rather than a direct causal relationship. We therefore adopt a more cautious treatment of its role. Specifically, we now retain only a limited discussion of two possible secondary influences of chlorine: (i) experimental results at relatively low pressures and temperatures (<0.5 GPa) indicating a positive correlation between Cl concentration and CaSO₄ solubility (Newton, 2004); and (ii) the stronger affinity of Cl for Fe relative to sulfur, which may indirectly affect sulfur transport by promoting metal–chloride complexation and influencing fluid–rock interaction processes. Accordingly, in the revised manuscript (Lines 288–314), we have softened statements regarding the role of chlorine in sulfur enrichment and limited the discussion to a concise and cautious interpretation. We hope that this revised and more cautious treatment adequately addresses the reviewer’s concern!

(4) I’m not sure who your flux models work – is this for a fictional subduction zone, or for a global S flux from the slab? It needs to be better explained.

Reply: We thank the reviewer for this constructive comment. We agree that the description of the sulfur flux calculations in the original manuscript was not sufficiently clear. In the revised manuscript, we have substantially clarified both the rationale underlying our flux estimates and the limitations associated with their extrapolation. As discussed above, the justification for extending results from the Sumdo system to global subduction zones can be summarized as follows.

(1) Representativeness of the Sumdo system. Although our study focuses on a single natural system, the Sumdo belt records a well-constrained cold subduction regime that represents a widespread and geodynamically significant class of subduction zones globally. While it does not encompass the full spectrum of global subduction variability (e.g., hot subduction systems), the Sumdo system provides a robust natural reference for sulfur cycling in cold subduction environments. As such, it provides a useful natural reference for the general assessment of sulfur cycling processes in subduction zones at the global scale, particularly within cold subduction settings that play a major role in volatile recycling.

(2) Direct constraints on sulfur concentrations in slab-derived fluids. Our study provides the first direct, in situ quantification of sulfur concentrations in natural slab-derived fluids released from the altered oceanic crust at sub-arc depths. Notably, the measured sulfur contents fall within the range predicted by previous experimental, thermodynamic, and arc geochemical studies. This consistency lends confidence to the use of our measured sulfur concentrations as a geologically realistic parameter for estimating sulfur fluxes transported by sub-arc fluids at the global scale.

(3) Constraints on H₂O flux as a key parameter in sulfur flux calculations. In calculating sulfur fluxes, the H₂O flux represents the other key parameter. In the original manuscript, we relied directly on literature estimates of H₂O fluxes at sub-arc depths (80–90 km). In the revised manuscript, to provide a more rigorous calculation, we performed thermodynamic calculations constrained by the geological and P–T conditions of the Sumdo system to estimate slab

dehydration extents at sub-arc depths. The results indicate that approximately 60% of the water initially stored in altered oceanic crust is released at depths of 80–90 km. Based on this dehydration fraction, and integrating average H₂O contents of altered oceanic crust, global subduction zone length, and crustal thickness (Alt et al., 1999; Stern et al., 2002; Ribeiro et al., 2026)(see revised Methods for details), we estimate a global H₂O flux of ~82.5 Mt/y released from subducted oceanic crust at sub-arc depths. This estimate closely matches independent literature constraints (~91 Mt /y, Li et al., 2020), supporting the robustness of our water-flux parameterization.

(4) Sulfur flux calculations in sub-arc fluids supported by previous independent studies. By combining the measured sulfur concentrations in slab-derived fluids with the calculated global H₂O fluxes released from the altered oceanic crust at-sub depths, we estimate the sulfur flux transported by sub-arc fluids and quantify the efficiency of slab-derived sulfur release at sub-arc depths, as well as its contribution to sulfur enrichment in arc systems. These estimates are quantitatively in agreement with independent experimental, theoretical, and geochemical constraints (Walters et al., 2020; Li et al., 2022; Taracsák et al., 2023), lending support to a cautious and reasonable extrapolation from the Sumdo system to global subduction zones.

We consider this extrapolation to be reasonable, while fully acknowledging the associated limitations. In the revised manuscript (Lines 331–360, Lines 539–550), we now provide a detailed discussion of both the rationale and the constraints involved in extending results from the Sumdo system to broader subduction zone settings. We sincerely hope that these revisions adequately address the reviewer’s concerns!

(5) Data tables are provided as a Word document, which is not particularly useful for other researchers who may want to use your data – you probably have an excel/csv table for these – please

Reply: We appreciate the reviewer for this helpful suggestion. We have converted original Table 1 and Table S1 from a Word document into Excel format and now provide them as Tables S1 and S2 in the Supplementary Information. In addition, we have expanded these tables to include standard deviations (SD) as well as minimum and maximum values.

Furthermore, we have added several new datasets to the Supplementary Information. These include (i) EPMA major element compositions of host omphacite across the investigated multiphase fluid inclusions (Table S3), and (ii) New thermodynamic modelling results constrained by the geological and P–T conditions of the Sumdo system, including calculated sulfur speciation in slab-derived fluids and estimates of slab dehydration extents (Table S4 and S5).

Detailed comment

Lines 47-48: While the 20-30% fluid contribution from the slab to the mantle wedge, and subsequently to the arc, is indeed the value in Li et al. (2020) you cite, but this is one of only few papers that argue for low amounts of S being transferred from the slab to the wedge, and there are many others that say the opposite. For example, in Central America, we estimated 40-70% sulfur addition to the mantle wedge from the slab in Taracsák et al. 2023. The way it is currently written implies there is only this 20-30% range in the literature, which is not the case.

Reply: We apologize that the original wording did not adequately reflect the full range of

published estimates for slab-derived sulfur transfer to the mantle wedge.

In the revised manuscript, we now explicitly acknowledge that estimates of slab-derived sulfur addition to arc magma sources span from ~20–30% to as high as ~40–70%, based on both thermodynamic modeling and geochemical constraints from natural arc systems (e.g., Li et al., 2020, 2021; de Moor et al., 2002; Muth and Wallace, 2022; Taracsák et al., 2023). We further clarify that this variability likely reflects differences among subduction zone settings, but also—perhaps more fundamentally—differences in the approaches used to quantify sulfur transfer, including theoretical modeling and mass-balance calculations based on arc volcanic rocks. We further clarify that this wide range likely reflects not only variability among subduction zone settings, but also fundamental differences in these indirect methodological approaches. Although these indirect approaches provide important constraints, they do not directly measure sulfur concentrations in slab-derived fluids and therefore may yield inconsistent conclusions regarding the contribution of slab-derived sulfur to arc magma budgets. In this context, we emphasize that our study provides direct observational constraints on sulfur concentrations in natural slab-derived fluids at sub-arc depths. These data offer an independent line of evidence that helps bridge the gap between model-based predictions and sulfur enrichments inferred from arc magmas.

Accordingly, we have revised the relevant text (Lines 54–63), which now reads: “However, mass-balance calculations indicate that such fluids contribute a highly variable fraction of arc-related sulfur emissions, exposing a fundamental mismatch between the observed arc outputs and the estimated slab-derived inputs¹⁸⁻²². For example, sulfur isotopic compositions of ultrahigh-pressure (UHP) metamorphic rocks, combined with thermodynamic modeling, suggest that slab-derived fluids may account for only ~20% of the sulfur budget of arc magmas^{22,23}. In contrast, sulfur enrichments recorded in melt inclusions from natural arc basalts imply substantially larger contributions, on the order of ~40–70%²⁰. This discrepancy may partly reflect the compositional diversity of slab-derived fluids among different subduction systems. More fundamentally, however, it likely arises from the lack of direct constraints on sulfur contents and fluxes in natural slab-derived fluids themselves. ”

Line 51: This implies that people argued for extra S in the mantle before subduction. I have not seen papers suggesting high S contents in the subarc mantle (relative to normal, depleted upper mantle) under arcs prior to subduction-derived S addition. Is there a reference for this?

Reply: We agree that the original wording was unclear. Our intention was to convey that, although slab-derived sulfur represents a major and essential source for arc magmatic systems, it may not always be sufficient on its own to account for the total sulfur released in arc settings. In such cases, additional contributions—such as mantle-inherited sulfur—may also play a role in the overall sulfur budget of arc magmas. To avoid ambiguity or potential misinterpretation, we have removed this statement in the revised manuscript.

Line 57: a minor issue but melt inclusion paper you cite here (Muth & Wallace 2021) used tephra samples, not lavas.

Reply: Thanks. We have corrected the text in the revised manuscript (Lines 65–69), which now reads: “To date, estimates of sulfur concentrations in subduction zone fluids have relied predominantly on indirect approaches, including analyses of sulfur concentrations and isotopic

compositions in exhumed metamorphic rocks, melt inclusions in arc basalts and tephra, high-temperature and high-pressure (HT–HP) experiments, and thermodynamic modeling^{18-22,24-29}.”

Line70-72: you cite Evans 2012 (a redox focussed review paper) and Bekaert et al. 2021 (a volatile cycle review paper), rather than papers that look at subducting S budgets (de Moor et al. 2022 Chem Geol, Peccia et al. 2025 Chem Geol), that are much more detailed and more relevant to your work. Crucially, both of these papers suggest it is the primary igneous crust that has the largest S budget followed by the AOC– and the total igneous package (AOC + primary) are the most important sources of S. Is there any proof that your eclogites represent AOC, or can they be metamorphosed primary crust (unaltered)?

Reply: We thank the reviewer for the insightful suggestion and for drawing our attention to these important references. These references have now been incorporated into the revised manuscript (Line 85).

Regarding the nature of the protolith of the Sumdo eclogites, previous work has constrained their origin based on whole-rock major and trace element systematics and Sr–Nd isotopic compositions (Yang et al., 2009; Liu et al., 2019a). As summarized in Fig. 1 (after Liu et al., 2019a), these geochemical data indicate that the Sumdo eclogites were derived predominantly from altered oceanic crust, with contributions from relatively fresh igneous components. This clarification has now been added in the revised manuscript (Lines 185–188).

Fig.1 (a) Chondrite-normalized rare earth element diagram for the Sumdo eclogite samples. (b) Plot of initial Sr–Nd isotopic compositions of the Sumdo eclogite. Figures are cited from Liu et al. (2019).

Line 99: the presence of pyrite is interesting. Looking at the recent model results of Beaudry & Sverjensky 2024 (G-cubed), who extended the Deep Earth Water model’s calibration range and

included oxidised S species, they suggest that under conditions similar to what you imply for these samples (~3GPa, 600-800 Celsius), pyrite is present at FMQ-1 to FMQ+2 (at higher T, it is more restricted to FMQ+0.5 to +1.5). Critically, their model results provide S contents in fluids in equilibrium with mafic eclogite that reach the S contents you propose based on your Fis (and even higher values), so it is a very relevant paper to your results, and you currently don't cite or integrate this into your paper. I think you need to carefully consider their findings regarding the S carrying capacity of fluids at higher fO_2 than those modelled in Li et al. 2020, as they predict up to an order of magnitude more S in the fluid.

Reply: We sincerely appreciate the reviewer for sharing this valuable reference. We have carefully examined the work of Beaudry and Sverjensky (2024). It is indeed a nice paper. Their findings provide strong independent support for our results. Specifically, (1) the sulfur concentrations predicted for slab-derived fluids at sub-arc conditions in Beaudry and Sverjensky (2024) overlap well with the sulfur contents quantified in our study (3.2–10.3 wt.%); (2) their thermodynamic modeling demonstrates that fluids equilibrated with pyrite at sub-arc depths can be dominated by oxidized sulfur species. This is fully consistent with our observations, including the presence of residual pyrite in the Sumdo eclogites (Fig. S2) and the inference of oxidized sulfur species in the initially entrapped fluids based on our DEW modelling (Fig. S7). We have now incorporated and discussed Beaudry and Sverjensky (2024) at several appropriate locations in the revised manuscript (Line 69, 73, 198, 208, 225, 233, 288–290). We thank the reviewer again for drawing our attention to this important and relevant contribution!

Line125-128: the reconstructed fluid compositions are very interesting, particularly with respect to their high Cu contents. This is because, as far as I'm aware, Cu has been suggested not to mobilise during slab devolatilisation (e.g. Jenner, 2017 Nature Geo). Your paper could be more broad interest I think if you explored other elements alongside S, such as Cu, H₂O or Cl. For example, to see if your FIs have enough H₂O to enrich the mantle wedge in H₂O and match estimates for mantle wedge H₂O contents that range from 0.3 to 1wt% in the Marianas (Kelley et al. 2010 J. Petrol). The 0.1% to 1% fluid addition in Fig 3A seems a bit too low for that. Also, how much Cu would be added to the mantle under the same circumstances, considering your Fis have 5 wt%, and the upper mantle only ~30 ppm?

Reply: We thank the reviewer for these excellent and constructive suggestions.

(1) We agree that H₂O fluxes in subduction zones may vary among different tectonic settings and that precise values are inherently uncertain. In the revised manuscript, we therefore incorporated thermodynamic modelling constrained by the geological and P–T conditions of the Sumdo system to quantify slab dehydration at sub-arc depths (~80–90 km). The results indicate that approximately 60% of slab-bound water is released at these depths (Fig. S6). Treating the Sumdo system as representative of altered oceanic crust dehydration in cold subduction zones, and integrating this dehydration fraction (~60%) with the average H₂O content of altered oceanic crust, global subduction zone length, and crustal thickness (Alt et al., 1999; Stern et al., 2002; Ribeiro et al., 2026), we estimate a global H₂O release flux of ~84.5 Mt/yr at ~80–90 km depth. This estimate is consistent with independent literature constraints (~91 Mt/yr, Li et al., 2020), supporting the robustness of our water-flux parameterization. Therefore, this approach provides an estimate of the average global H₂O flux from the altered

oceanic crust at sub-arc depths, while we explicitly acknowledge that the precise magnitude of H₂O flux may vary among individual subduction systems due to differences in tectonic and thermal conditions.

(2) Furthermore, we agree that the reconstructed fluid compositions have implications extending sulfur, particularly in light of the unexpectedly high Cu concentrations measured in the inclusions. Given the close geochemical relationship between sulfur and copper (Richards, 2015; Farsang and Zajacz, 2025), we have added a concise discussion in the revised manuscript (Lines 391–422) to explore the implications of slab-derived sulfur-rich, oxidized fluids for Cu mobility in the subduction zone. This new discussion focuses on two main aspects:

(1) We acknowledge that Cu mobility during slab devolatilization remains debated. Some studies argue that Cu is largely retained in the slab and is not efficiently transferred by slab-derived fluids (e.g., Jenner, 2017; Zou et al., 2024; Ren et al., 2025), whereas other work suggests that Cu can be mobilized under specific conditions, particularly when complexed by sulfur-bearing fluids (e.g., Li et al., 2013; Walters et al., 2021). Our fluid inclusion data provide direct natural evidence that slab-derived fluids at sub-arc depths can contain elevated Cu concentrations. In the revised manuscript, we therefore emphasize that the combination of sulfur-rich and oxidized fluid compositions provides a plausible mechanism for Cu mobilization in subduction zones.

(2) We further address the apparent discrepancy between Cu-rich slab-derived fluids and the lack of systematic Cu enrichment in many arc magmas. Many geochemical signatures of many arc basalts indicate that their mantle sources are not systematically enriched in copper, implying limited net addition of slab-derived copper to the arc magma source (Muth et al., 2022; Lee et al., 2012; Lee and Tang, 2020; Zhao et al., 2022). In contrast, some metasomatized sub-arc xenoliths show elevated copper compared to the depleted mantle compositions (McInnes et al., 1999; Kepezhinskis et al., 2002; Lorand et al., 2013; Tassara et al., 2018). In the revised manuscript, to reconcile this apparent mismatch, we propose three possible processes. (i) Slab-derived Cu may be efficiently sequestered or “filtered” during fluid–rock interaction at the slab–mantle interface or within the mantle wedge, for example, through sulfide precipitation. Consistent with this interpretation, Cu enrichment has been found at eclogite–peridotite interfaces (representing slab–mantle boundary), including in CCSD samples reported in our recent study (Tan et al., 2025, GRL), where Cu concentrations at the interface are significantly higher than in adjacent eclogite and peridotite. We note that the previous study reported Cu concentrations for the interface and adjacent lithologies only; a detailed mechanistic investigation of Cu enrichment at these interfaces is beyond the scope of the previous study (Tan et al., 2025) and will be addressed in future work. (ii) Copper transport may be spatially heterogeneous, with focused fluid flow through channels, leading to localized Cu enrichment within the mantle wedge rather than uniform distribution. During ascent and interaction with surrounding mantle peridotite, Cu may be preferentially unloaded via sulfide formation, limiting its transfer to arc magmas. Sulfide saturation and magmatic differentiation processes may buffer Cu concentrations in arc melts, thereby reducing contrasts between MORB and arc lavas despite variable slab-derived Cu inputs (Walters et al., 2021).

Taken together, these considerations indicate that Cu enrichment in arc magmas and the formation of copper deposits are governed by a combination of factors, including the capacity of slab-derived fluids to mobilize copper, the influence of sulfide sequestration during fluid

transport, and subsequent magmatic differentiation processes. Nevertheless, our results demonstrate that sulfur-rich, oxidized sub-arc fluids are capable of effectively mobilizing Cu from the subducted slab. This provides a necessary material basis for copper transfer into the mantle wedge and, ultimately, for copper enrichment in arc magmas, highlighting the potential role of slab-derived fluids in the genesis of porphyry copper deposits in arc settings.

We thank the reviewer again for encouraging us to broaden the scope of the discussion. We believe that these revisions improve the overall significance and appeal of the manuscript to a broad readership!

Line 136: I have some issue with your interpretation of the FIs being primary. In line 110, you mention FIs are parallel with the C axis (a cleavage direction for cpx). You also mention they are confined to cores and have negative crystal morphologies. But together, these observations cannot rule out that the inclusions are in fact secondary, entrapped in cracks within the omphacite after formation, followed by further crystal growth along the rims. Negative crystal morphologies alone cannot be used to argue for a primary origin (see Goldstein 2003 review on FI petrography). In any case, you need to provide some further arguments why your inclusions are representative for peak metamorphic conditions, considering a potential secondary origin relative to omphacite cores.

Reply: We appreciate the reviewer for the comments on the origin of fluid inclusion.

(1) In the revised manuscript (Lines 125–142), we have further clarified this issue and now present several independent lines of evidence that collectively support a primary origin for the fluid inclusions. (i) The fluid inclusions are consistently confined to the cores of individual omphacite grains and do not occur as planar trails or arrays that crosscut the host crystal. Secondary inclusions are typically associated with healed fractures or crack-related inclusion trails; however, detailed petrographic examination reveals no evidence for healed cracks or microfractures in the host omphacite that could have acted as pathways for secondary entrapment (Fig. S2). (ii) The inclusions exhibit remarkably uniform phase assemblages and similar filling degrees across multiple grains. In contrast, secondary inclusions commonly display greater variability in phase proportions and compositions due to episodic trapping along fractures. (iii) The inclusions display elongate, negative-crystal-like morphologies that mimic the crystallographic habit of the host omphacite. This negative-crystal-like geometry is consistent with primary entrapment and provides an additional line of evidence for a primary origin, although it is not, by itself, a definitive criterion (Goldstein, 2001). Taken together, we consider all the criteria and interpret these lines of evidence as most consistent with primary entrapment during omphacite growth.

(2) Regarding the stage and P–T conditions of the fluid inclusions hosted in omphacite, previous studies have tightly constrained the P–T conditions of omphacite growth in the Sumdo eclogites (Table below, Liu et al., 2019b). This study suggested that omphacite crystallization was restricted to the peak metamorphic stage. Accordingly, we interpret the omphacite-hosted fluid inclusions to record fluids trapped during peak to near-peak UHP metamorphic conditions, corresponding to sub-arc depths.

Table 3
Mineral parageneses in 4 stages of metamorphism.

Minerals	Prograde stage	Peak stage	Retrograde I stage	Retrograde II stage
Garnet	Grt I (Gro ₃₀₋₃₄ PyP ₁₈₋₁₂)	Grt II ₁ (Gro ₃₅ PyP ₁₇)	Grt II ₂ (Gro ₃₄₋₂₆ PyP ₂₁₋₂₈)	–
Omphacite	–	Omphacite Jd ₃₀₋₃₈	–	–
Amphibole	Amp I (FeO 9–10%)	Amp II FeO 9–13%	Amp III (FeO 7–12%)	Amp IV (FeO 6–8%)
Rutile	Rt in core of Grt.I	Rt in omphacite	Rt in matrix	–
Phengite	–	Ph (3.6Si pfu)	Ph (3.1–3.2Si pfu)	–
Epidote_group	Czo-I in core of Grt-I	–	Ep-II (Fe pfu 0.32–0.47)	Ep-III (Fe pfu 0.08–0.15), Zo IV
Quartz	Qtz in core of Grt-I	Coesite pseudomorph??	–	Qtz in the matrix

This table summarizes the growth stages of various metamorphic minerals, showing that omphacite occurs exclusively at the peak metamorphic stage. Table is from Liu et al. (2019b).

Line 140: I doubt that you can rule out post entrapment re-equilibration. H⁺ diffuses rapidly within cpx, so you could gain or lose it depending on H activity around the grains. Fe²⁺ can also diffuse in and out. Both of these elements could influence the redox within your inclusions. This leads me to think that the S may have been oxidised in the fluid originally, not reduced, and the chalcopyrite formed due to post-entrapment processes. Can you rule this out, and if yes, how? If not, this needs to be considered.

Reply: Many thanks for this constructive comment. As noted by the reviewer, H⁺ can diffuse relatively rapidly within clinopyroxene, and Fe²⁺–Fe³⁺ exchange between the host omphacite and the inclusions is also possible. Both processes could modify the redox state within fluid inclusions after entrapment.

With respect to H diffusion, this process is difficult to directly constrain in natural samples, and we acknowledge that it cannot be excluded in this study (Lines 241–243). Regarding Fe exchange, we have added new EPMA profiles across host omphacite surrounding the representative multiphase fluid inclusions. These data show that omphacite in the immediate vicinity of inclusions contains slightly lower FeO contents (0.3 wt.% variations) compared to regions farther from the inclusions (Fig. S6). Such Fe content variations could plausibly have modified the redox state within the inclusions after entrapment.

To further constrain the primary sulfur speciation in the entapped fluids, we performed thermodynamic modeling using the DEW framework, constrained by the geological and P–T conditions of the Sumdo system. These calculations indicate that sulfur in slab-derived fluids at sub-arc depths is dominantly present as oxidized species (e.g., SO₄²⁻ and HSO₄⁻) (Fig. S7). On this basis, we now interpret the entrapped fluids as being most likely oxidized at the time of entrapment at-sub depths.

We therefore revise our interpretation to explicitly consider that post-entrapment processes, including Fe exchange between the host omphacite and the inclusions, may have driven localized redox reactions that partially reduced sulfur species and promoted chalcopyrite precipitation within the inclusions. Alternatively, hydrogen diffusion across the inclusion–host boundary may also have contributed to such redox modification, although its quantitative effect remains difficult to assess. This clarification regarding sulfur speciation has been incorporated in the revised manuscript (Lines 226–253).

Accordingly, in the revised manuscript, we clarify that the presence of reduced sulfur phases within the inclusions does not necessarily reflect the primary redox state of the slab-derived fluids, but may instead record post-entrapment modification of initially oxidized, sulfur-rich fluids. Furthermore, we also note that although post-entrapment modification may affect sulfur speciation within the inclusions, it is unlikely to significantly reduce the measured sulfur

concentrations, given the limited sulfur content of the host omphacite. If anything, such post-entrapment processes would tend to elevate the preserved sulfur content relative to the original trapped fluid. Therefore, while we revise our interpretation of sulfur speciation, this does not weaken our conclusions regarding high sulfur contents in slab-derived fluids; instead, it further supports the robustness and importance of our findings.

Line145-146: I don't understand this sentence. What is replaced? Feels like the subject of the sentence is not defined.

Reply: Thanks. Our original intention was to convey that sulfides occurring within the fluid inclusions are chalcopyrite, whereas sulfides in the host eclogite are dominated by pyrite, thereby excluding mechanical transfer of sulfides from the host into the inclusions. To avoid any ambiguity or confusion, we have removed this sentence from the revised manuscript.

Line157: you say that your S contents at 3-10 wt% is similar to the literature range, but in fact it is narrower by an order of magnitude, and you exclude the lower end. So rather than saying it is similar, I would say it is more restricted.

Reply: Thanks for this helpful suggestion. The revised text (Line 196–205) now reads: Furthermore, the sulfur concentrations measured in this study (~3.2–10.3 wt.%, Table S2) fall within the broader range (~0.5–15 wt.%) inferred for subduction zone fluids from thermodynamic models and high-pressure experiments^{22,24-29}. The wide variability predicted by these indirect approaches likely reflects uncertainties in pressure–temperature–redox conditions, fluid-rock interactions, and mineral equilibria. In contrast, our analysis of primary multiphase fluid inclusions avoids many of these limitations and provides a direct quantitative constraint on the sulfur concentrations in slab-derived fluids at sub-arc depths. These measurements not only validate previous experimental and theoretical predictions of elevated sulfur contents in sub-arc fluids but also provide more restricted natural constraints on the range of sulfur concentrations in slab-derived fluids.”

Line160-161: Well, you argue that your results are more faithful than thermodynamic models to the real fluid composition. But surely, your one sample is restricted to one P-T-X path, so hardly a representation of global subduction zone variability that include hot and cold, steep and shallow, and old and young slabs (see e.g. Syracuse et al. 2010). The restricted S content range restricted range may be representative of the fact that you only look at a single P-T-X.

Reply: We thanks the reviewer for this comment and fully acknowledge the variability among subduction zones, such as differences between hot and cold subduction regimes. Accordingly, sulfur contents in slab-derived fluids are expected to vary among different subduction settings. In our dataset, sulfur concentrations measured in the fluid inclusions also show variability, ranging from 3.2 to 10.3 wt.% (Table S2). We suggest that this range may, at least in part, overlap with sulfur contents characteristic of fluids generated in different subduction environments. To account for this variability in subsequent discussion and modelling, we therefore use the mean sulfur concentration (6.3 wt.%) together with its associated standard deviation (SD = 1.8 wt.%) as representative values in our calculations.

Furthermore, we also recognize that extrapolating sulfur concentrations from the Sumdo system to global subduction zones involves inherent limitations. Nevertheless, the Sumdo belt

represents a well-characterized cold subduction regime, which constitutes a widespread and geodynamically significant class of subduction zones. On this basis, we suggest that sulfur concentrations comparable to those documented here may also occur in similar cold subduction environments, although confirmation from additional natural systems will require future investigation. Importantly, when using the average sulfur content of sub-arc fluids (~6 wt.%) in our flux calculations, the resulting estimates of sulfur release efficiency and contributions to arc systems are quantitatively consistent with previous indirect constraints (Walters et al., 2020; Li et al., 2022; Taracsák et al., 2023). This agreement suggests that the sulfur characteristics observed in the Sumdo system are unlikely to be anomalous, but may instead reflect a broader feature of sulfur-rich slab-derived fluids in comparable subduction settings. These points have been clarified in the revised manuscript (Lines 205–213, 331–360). We sincerely hope that this explanation satisfactorily addresses the reviewer’s concern!

Lines 164-167: While I very much agree with this statement, you kind of put this here in a vacuum, and somewhat feels like you are implying this is a new idea (and cite no other paper that suggested this). In fact, lots of papers argued for extensive S transfer from slab to mantle wedge, both old and new (Métrich et al. 1999 EPSL, de Hoog et al. 2001 EPSL, Walters et al. 2020 GPL, Chowdhury & Dasgupta 2019 Chem Geol, Beaudry & Sverjensky 2024 G-cubed, Muth and Wallace 2022 EPSL, Taracsák et al. 2023 EPLS, Kawaguchi et al. 2022 JPetro, just to list a few). Yes, your results confirm this, and are valuable, but they add to existing evidence that you should mention here.

Reply: We appreciate the reviewer for this important and well-taken suggestion. We fully agree that extensive sulfur transfer from the slab to the mantle wedge has been proposed and discussed in numerous previous studies, and our original wording did not sufficiently acknowledge this body of work. In the original manuscript, our intention was not to imply that efficient slab-to-wedge sulfur transfer is a new concept. Rather, we aimed to emphasize that sulfur-rich fluids recorded in HP–UHP metamorphic rocks and associated veins are not unique to the Sumdo eclogites, but have been documented in multiple subduction settings worldwide (Phillippot et al., 1991; Svensen et al., 1999, 2001; Frezzotti and Ferrando, 2015). What distinguishes our study is that it provides the first direct, in situ quantification of sulfur concentrations in slab-derived fluids at sub-arc depths, thereby filling a critical observational gap between arc magma-based inferences and slab dehydration processes.

The revised text (Lines 205–213) now reads: “Importantly, sulfur-rich slab-derived fluids are not unique to the Sumdo system. Numerous studies based on observations of arc magmas and melt inclusions, together with experimental constraints and thermodynamic modeling, have demonstrated extensive sulfur transfer from the subducting slab to the mantle wedge, highlighting the critical role of slab-derived fluids^{11-21,27}. Moreover, sulfide- and sulfate-bearing fluid inclusions have been documented in HP–UHP eclogites and associated veins from other orogenic belts worldwide, such as the Western Alps and the Caledonides^{32,41-43}. Collectively, these observations indicate that sulfur enrichment in slab-derived fluids at sub-arc depths is likely a common feature of deep subduction zones rather than an isolated characteristic of the Sumdo system.”

Line 169: I think this subsection title misses a word, or mechanism is not the best way to put it.

“Formation of ...” may be better?

Reply: Thanks, we have revised it (Line 215).

Line173: reference 33 is in fact a review of a paper in Nature Geo, not the paper itself that presents the results – that would be Farsang and Zajacz 2024 Nature Geo.

Reply: We appreciate this suggestion and have now cited the original study by Farsang and Zajacz (2024, Nature Geoscience) instead of the review article (Line 393).

Line176: again, oxidised species are not considered here, yet these can carry more S than reduced species – see earlier comments.

Reply: We thanks the reviewer for this insightful comment. We have revised the manuscript to fully incorporate oxidized sulfur species in the discussion. The revised text (Lines 217–225) now reads: “In altered oceanic crust, sulfur is initially hosted in both sulfate- and sulfide-phases, which are progressively destabilized with increasing depth during subduction^{43,44}. Some studies suggest that sulfate-bearing minerals mostly decompose and release oxidized sulfur species (SO_4^{2-}) at relatively shallow forearc depths (<70 km), whereas sulfide phases remain stable to greater depths and break down under eclogite-facies conditions (>70 km), liberating reduced sulfur (e.g., S^{2-} , HS^-) into coexisting fluids^{22,23,43,45}. However, more recent experimental, thermodynamic, and geochemical studies indicate that oxidized sulfur species may remain dominant even at sub-arc depths, despite increasing pressure and temperature^{18-21,26,27}.”

Lines178-180: I already mentioned this earlier; but pyrite is not an indicator of low $f\text{O}_2$ (pyrrhotite would be), and in fact a higher $f\text{O}_2$ may be needed for pyrite. Can you be sure that the cpy is not formed by post entrapment reduction of S, considering Fe and S can readily exchange electrons? Omphacite can contain aegirine and Fe^{3+} - in fact, Weller et al. 2016 measured a $\text{Fe}^{3+}/\text{Fe(T)}$ of 0.33-0.45 in peak omphacite, so there is plenty of it to reduce the little S (with respect to total mass) present in the FIs. And a similar argument can be made for calcite – can the Ca scavenged from the omphacite (it has 18 wt% CaO)? A similar process is described for opx-hosted, CO_2 -rich FIs from mantle xenoliths (Berkesi et al. 2012 EPSL), where reactions between the fluid and the host formed carbonates.

Reply: We are grateful to the reviewer for this important and detailed comment. In the revised manuscript, we have fully reconsidered the possibility of post-entrapment modification and re-evaluated the sulfur speciation of the fluids initially trapped at sub-arc depths.

As noted by the reviewer, pyrite is not an indicator of low oxygen fugacity, and its presence in the host eclogite does not require reducing conditions. We therefore do not use pyrite as evidence for reduced sulfur speciation at the time of fluid entrapment. Instead, thermodynamic modeling constrained by the P–T conditions of the Sumdo system indicates that slab-derived fluids at sub-arc depths are dominantly characterized by oxidized sulfur species (SO_4^{2-} and HSO_4^-) at the time of entrapment (Fig. S7).

To assess the potential for post-entrapment modification, we performed quantitative EPMA profiling of host omphacite across representative exposed fluid inclusions. These analyses reveal limited but measurable compositional gradients, with maximum variations of ~0.4 wt.% FeO and ~0.6 wt.% CaO between regions adjacent to inclusions and those farther away (Table S3, Fig. S6). These results indicate that post-entrapment element exchange between the trapped

fluids and the host omphacite did occur, albeit to a limited extent. Accordingly, we now explicitly acknowledge that Fe exchange between the entrapped fluids and Fe³⁺-bearing omphacite could have promoted localized redox reactions after entrapment. Such reactions may have partially reduced initially oxidized sulfur species and led to the precipitation of chalcopyrite within the fluid inclusions.

Finally, we emphasize that although post-entrapment fluid–host interactions may have modified sulfur speciation, they do not affect our primary conclusion regarding sulfur concentrations. Sulfur is absent in the hosted omphacite, and therefore, post-entrapment exchange processes cannot significantly modify the total sulfur concentrations recorded by the fluid inclusions.

Line228-230: if I understand correctly, you argue that sulfur is more mobile because of the high Cl content of fluids, and that there is a coupling effect. Yet, there is plenty of evidence that S is fairly fluid mobile as it is, especially under oxidising conditions. Cl is always very fluid mobile (elevated in all arcs) – so this can simply be the case of two fluid mobile elements behaving similarly, because they have similar affinity to fluids under certain conditions? The citation you provide in line 227 does not add to your argument – it is a melt inclusions paper from 22 years ago, that highlights there is a lot for Cl and S in arc melt inclusions, and they modelled a fluid composition based on mass balance – it is not an observed value for fluid S and Cl content.

Reply: We appreciate the reviewer’s constructive comment and agree that the original manuscript overemphasized the role of chlorine in enhancing sulfur concentrations in slab-derived fluids. In response, we have made the two revisions as follows:

(1) Guided by recent thermodynamic results (Beaudry & Sverjensky, 2024), we now focus on the role of major cations—particularly Ca and Na—which exert a stronger and more quantifiable control on sulfate solubility and sulfur transport under UHP conditions. This interpretation is more directly supported by our observations of elevated CaO (10.16 wt.%) and Na (3.22 wt.%) in association with exceptionally high sulfur contents in the Sumdo fluid inclusions.

(2) Previous studies have also suggested a potential role for chlorine in promoting sulfur enrichment in subduction zone fluids (Newton, 2004). In addition, chlorine may also exert an indirect influence through its stronger affinity for iron relative to sulfur, potentially favoring Fe–Cl complexation and thereby suppressing sulfide reprecipitation during fluid transport. Such a mechanism could facilitate longer-distance sulfur migration within the mantle wedge. The elevated concentrations of chlorine (3.5–6.4 wt.%), iron (2.80–8.98 wt.%), and high sulfur observed in the Sumdo fluid inclusions are broadly consistent with this possibility. Nevertheless, we also note that whether the coupled enrichment of these two fluid-mobile elements, chlorine and sulfur, reflects similar behavior under sub-arc conditions—and the extent to which chlorine enhances sulfur transport—remains to be further investigated. Overall, in the revised manuscript (Lines 288–314), we have weakened the role of Cl in enhancing sulfur concentrations in sub-arc fluids. We hope that this revised and more cautious treatment adequately addresses the reviewer’s concern!

Line240-242: Even for one arc, these estimates vary a lot – see de Moor et al 2022 for central America, who estimate that between 10-60% of S is lost to the mantle wedge from the slab. So, most S may enter the deep mantle, but also, most may go into the arc.

Reply: We thank the reviewer for this important comment. We have revised the manuscript accordingly (Lines 319–323). The revised text now reads: “Previous sulfur mass-balance estimates for subduction zones indicate that the fate of subducted sulfur is highly variable, with sulfur either being transferred into the mantle wedge or retained within the slab and transported into the deeper mantle^{6,8,10,18-20,26}. For example, studies of Central American arc basalts suggest that ~10 to 60% of slab-derived sulfur may be incorporated into the mantle wedge¹⁸.”

Lines 248-271: the flux calculations are confusing because you do not actually explain what you want to show, and how you do it; are these values global fluxes, or just for a specific subduction zone? Is this for a certain thickness of crust. It feels a bit confusing.

Reply: We thank the reviewer for pointing out the lack of clarity in this section. We agree that the original presentation of the flux calculations was insufficiently explained. In the revised manuscript, we have therefore restructured this section to clearly define what is being calculated, at what scale, and how the calculations are performed. The aim of the flux calculations is to provide a cautious extrapolation from the Sumdo system to broader subduction zone contexts. This extrapolation is based on several reasonable considerations.

(1) The Sumdo eclogites record a cold subduction regime, representing a widespread and geodynamically significant class of subduction zones globally. Accordingly, the sulfur systematics documented here are expected to be broadly applicable to other cold subduction settings, while recognizing that they do not encompass the full diversity of global subduction environments.

(2) The investigated samples record fluid compositions at sub-arc depths, representing a critical window of slab-derived sulfur release. As discussed above, sulfur-rich sub-arc fluids are not unique to the Sumdo system but have been documented in multiple subduction settings worldwide. Furthermore, our study not only validates previous experimental and theoretical predictions of elevated sulfur contents in sub-arc fluids but also provides more restricted natural constraints on the range of sulfur concentrations in slab-derived fluids. As such, we suggest that our measured sulfur concentrations can serve as an important and reasonable parameter for sulfur flux calculations.

(3) To further strengthen the calculation, we re-evaluated the second key parameter controlling sulfur fluxes, namely the global H₂O flux released from the altered oceanic crust. We performed thermodynamic calculations constrained by the geological and P–T conditions of the Sumdo system to estimate dehydration. The results indicate that approximately 60% of the water initially stored in altered oceanic crust is released at depths of 80–90 km. Based on this dehydration fraction, and integrating average H₂O contents of altered oceanic crust, global subduction zone length, and crustal thickness (Alt et al., 1999; Stern et al., 2020; Ribeiro et al., 2026), we estimate a global H₂O flux of ~82.5 Mt/y released from subducted oceanic crust at sub-arc depths. This estimate closely matches independent literature constraints (~91 Mt /y, Li et al., 2020), providing confidence in the adopted parameterization.

(4) Finally, based on these parameters, we calculate sulfur fluxes carried by sub-arc fluids and further estimate the efficiency of sulfur release from the slab at sub-arc depths, as well as its contribution to sulfur enrichment in arc systems. Importantly, these estimates are quantitatively consistent with independent constraints derived from previous experimental, theoretical, and geochemical studies (Walters et al., 2020; Li et al., 2022; Taracsák et al., 2023), providing

additional support for the reasonableness and reliability of extrapolating our results from the Sumdo system to global subduction zones.

We explicitly acknowledge in the revised manuscript that sulfur fluxes may vary among different subduction zones and even within a single subduction system. Accordingly, our calculations are not intended to represent precise values for all subduction zones, but rather to provide a general estimate of sulfur cycling at sub-arc depths. We believe that our study can provide a good case for future comparative studies of sulfur cycling across diverse subduction environments. The above clarifications have been clearly indicated in the manuscript (Lines 331–390).

Line275: the paper you cite on sediment melt S solubility says 1800 ppm at the SCAS – which is not that high (similar to a basalt at the SCSS) – combine this with the fact that sediments rich in S are usually reduced (e.g. Peccia et al. 2025 Chem GEol), makes me question if sediments melts can in fact carry substantial amounts of S as you state.

Reply: Thanks. In the revised manuscript (Lines 332–335), we have softened this statement. Now it reads: “In unusually hot subduction systems, slab-derived melts have been proposed as important sulfur carriers, with sulfur concentrations reaching up to several thousand ppm⁵⁵⁻⁵⁹. However, these concentrations remain substantially lower than those measured in slab-derived fluids in this study.”

Line318: I cannot find this water flux anywhere in the referenced paper. You should really explain what this “flux” is, not just say that this is the water flux at 90 km depth (which means not much – every subduction zone will have a different H₂O flux due to different P-T-X and crustal thickness). Is this for a certain thickness of crust dehydrating?

Reply: We thank the reviewer for this insightful comment. We now explicitly clarify the calculation of global H₂O flux release from altered oceanic crust at sub-arc depths of 80–90 km. In the revised manuscript, we performed new thermodynamic modelling constrained by the geological and P–T conditions of the Sumdo system to quantify slab dehydration extents. These calculations indicate that ~60% of the water initially stored in altered oceanic crust is released at sub-arc depths. Treating the Sumdo system as representative of altered oceanic crust dehydration in cold subduction zones, we combined this dehydration fraction (~60%) with average H₂O contents of altered oceanic crust, global subduction zone length, and altered crustal thickness (Alt et al., 1999; Stern et al., 2020; Ribeiro et al., 2026). On this basis, we estimate the global H₂O release flux of ~84.5 Mt/yr from altered oceanic crust at sub-arc depths. This value closely matches independent global estimates of ~91 Mt/y at ~90 km depth (reported in the caption of Fig. 6 in Li et al., 2020), supporting the robustness of our water-flux parameterization. These clarifications have been incorporated into the revised manuscript (Lines 331–360, 538–550).

Line329: Same question applies for the sulfur input flux – what does this value represents?

Reply: We thank the reviewer for this clarification request. In the revised manuscript, we now explicitly clarify what the sulfur input flux represents and how it is used in our study.

Our goal is to place the Sumdo-derived observations within a broader framework for assessing sulfur cycling in subduction zones. To this end, we calculate the global sulfur flux transported

by sub-arc fluids derived from altered oceanic crust based on three key considerations: (i) the Sumdo eclogites record a cold subduction regime, which represents a widespread and geodynamically important class of subduction zones globally; (ii) sulfur-rich sub-arc fluids have been documented in multiple subduction settings and are not unique to the Sumdo system; and (iii) the global H₂O fluxes at sub-arc depths (~80–90 km) from altered oceanic crust adopted in our calculations are constrained by thermodynamic modeling and are consistent with independent estimates reported in the literature.

To evaluate the significance of the calculated sulfur flux at sub-arc fluids, we compare it with the initial sulfur input flux carried by subducted altered oceanic crust, estimated to be ~48.09 Mt/yr (Evans, 2012). On this basis, we estimate a sulfur release efficiency of $25 \pm 7\%$ at sub-arc depths. This value is broadly consistent with previous theoretical and modeling-based estimates (~45%, Walters et al., 2020). This agreement supports the internal consistency of our sulfur flux calculations as well as the reasonableness of the adopted sulfur input flux from altered oceanic crust. These clarifications have now been incorporated into the revised manuscript (Lines 331–360).

Line364-365: This value does not significantly exceed what was suggested for Central America (between 40-70% of S being slab-derived in arcs) in Taracsák et al. 2023 – so in a sense your results confirm what was already suggested based on arc melt inclusion data.

Reply: We thank the reviewer for this insightful comment. By explicitly incorporating the analytical uncertainty (SD = 1.8 wt. %) of concentrations into our calculations, we now estimate that slab-derived fluids contribute $70 \pm 20\%$ of the sulfur released in arc systems. We have revised the manuscript accordingly (Lines 600–602). The sentence now reads: “The calculated range overlaps well with previous estimates of ~40–70% derived from indirect constraints based on natural arc basalts²⁰, supporting the consistency and geological plausibility of our flux-based assessment.”

Methods section: This is very little detail considering these Raman analyses are the base of the paper - what peaks/spectral features were used to identify phases? How sure can we be that each phase is a pure mineral phase and not something similar, yet compositionally different? (e.g. cpy vs other sulfides, calcite vs dolomite or magnesite?). While I appreciate that you provide four representative spectra in the supplementary, I think you should give a more detailed summary here.

Reply: We thank the reviewer for this insightful suggestion. In the revised manuscript, we have added a detailed description of the criteria used to identify daughter minerals and fluid phases based on their diagnostic Raman spectral features, drawing on established Raman databases (<https://rruff.info/>) and previous work (e.g., Frezzotti et al., 2012).

(1) For sulfide phases, chalcopyrite (CuFeS₂) is characterized by a strong Raman peak at ~293 cm⁻¹, whereas other common sulfides display distinct and non-overlapping diagnostic peaks, including pyrite (FeS₂) at ~428 cm⁻¹, marcasite (FeS₂) at ~324 cm⁻¹, sphalerite (ZnS) at ~349 cm⁻¹, and galena (PbS) at ~136 cm⁻¹. In this study, the dominant sulfide Raman peak consistently occurs at ~292–293 cm⁻¹ (Fig. S3), with no additional peaks indicative of other sulfide species, supporting the identification of chalcopyrite as the sole sulfide phase within the

inclusions.

(2) For carbonate phases, calcite is characterized by strong Raman bands at ~ 284 and ~ 1086 cm^{-1} , whereas dolomite shows peaks at ~ 299 and ~ 1097 cm^{-1} , and magnesite at ~ 329 and ~ 1094 cm^{-1} . The carbonate daughter minerals in the Sumdo inclusions consistently exhibit a peak combination at ~ 283 – 284 and ~ 1086 cm^{-1} (Fig. S3), which is diagnostic of calcite and clearly distinct from other carbonate species.

(3) The aqueous fluid phase is identified by a broad Raman band between ~ 3300 and 3600 cm^{-1} (Fig. S3), corresponding to O–H stretching vibrations, indicating the presence of H_2O -rich fluid. Host omphacite is identified by its characteristic Raman peaks near ~ 670 and ~ 1012 cm^{-1} ; in our analyses, peak positions at ~ 672 and ~ 1015 cm^{-1} are consistently observed (Fig. S3), confirming the host mineral identity. Furthermore, no solid NaCl daughter minerals were detected in the fluid inclusions (which would be expected to show a strong Raman peak near ~ 358 cm^{-1}). We therefore infer that NaCl is present as a dissolved component in the aqueous fluid phase. Consistent with our previous work (Liu et al., 2019b), microthermometric constraints indicate salinities of ~ 10 – 22 wt.% NaCl equivalent, with an average value of ~ 15 wt.% adopted for bulk fluid composition calculations. These clarifications have been incorporated in the Method section of the revised manuscript (Lines 434–453).

Fig. S3 in the supplementary information. Representative Raman spectra of daughter minerals and fluid phases in multiphase fluid inclusions. Raman spectra of chalcopyrite, calcite, and aqueous fluid phases within multiphase inclusions hosted in omphacite from the Sumdo eclogite.

Line 535: “Approximately this sulfur is eventually” sounds strange, I suggest rewording this sentence.

Reply: Thanks. We have rephrased the sentence (Lines 879–881) accordingly. It now reads: This sulfur is ultimately returned to the surface via arc volcanism, accounting for ~ 70 % of the global sulfur output flux of ~ 16.8 Mt/y.”

Figure 2: It would be interesting to add a ternary plot with water, cpy, and cc as the three endmembers – if the inclusions have a common origin, the fractions of the three phases should be similar across all inclusions. Maybe this can go in as a supplementary figure.

Reply: We thank the reviewer for this helpful suggestion. In response, we have added two ternary diagrams to the supplementary materials (Fig. S5). The first diagram (A) illustrates the relative phase proportions of the inclusions, with the three endmembers defined as brine (H₂O + dissolved NaCl), calcite, and chalcopyrite. The second diagram (B) shows the reconstructed bulk compositions of the inclusions, with endmembers defined as H₂O + NaCl, CuFeS₂, and CaCO₃. As shown in these diagrams, both the phase proportions and bulk chemical compositions of the fluid inclusions cluster tightly, indicating a high degree of compositional uniformity among the inclusions. This consistency supports a common origin for the trapped fluids.

Fig. S5 in the supplementary information. Phase proportions and reconstructed bulk compositions of multiphase fluid inclusions.

Table 1: you could add standard deviation and minimum/maximum values, as you use those in your Fig. 3.

Reply: Thanks. We have revised Table 1 to include the standard deviations (SD) as well as the minimum and maximum values for each quantified component. For sulfur, concentrations range from 3.2 to 10.3 wt.%, with an average value of 6.3 wt.% and a standard deviation of 1.8 wt.%. In addition, following the reviewer's earlier suggestion, Table 1 and Table S1 in the original manuscript have been converted into an Excel-format dataset, and the complete data are now provided in Tables S1 and S2.

I hope these comments are helpful and will assist improving your ms.

Reply: We once again thank Dr. Zoltán Taracsák for the detailed and constructive comments and thoughtful suggestions, which have significantly improved the clarity, rigor, and overall quality of the manuscript!

Reviewer 2's comment

This manuscript presents the first in situ quantification of sulfur contents in natural slab-derived fluids from sub-arc depths, based on multiphase fluid inclusions hosted in omphacite from UHP eclogites in the Sumdo belt, southern Tibet. The authors convincingly demonstrate that these fluids contain exceptionally high sulfur concentrations (~6 wt.%), potentially accounting for up to 75% of the global arc volcanic sulfur output. This is an important contribution to understanding sulfur cycling agents and fluxes in subduction zones, and the study is timely, original, and methodologically sound. The dataset is of high quality, the interpretations are generally well supported, and the implications are broadly relevant to geochemistry, subduction dynamics, and volatile transfer processes. Overall, the manuscript is suitable for publication in Nature Communications after moderate revisions in order to improving methodological transparency and quantitative rigor.

Reply: We sincerely thank the reviewer for the positive evaluation of our manuscript. We greatly appreciate the constructive comments and insightful suggestions, which have helped to further improve the clarity and quality of our study. In response, we have carefully addressed all points raised by the reviewer. All revisions are clearly indicated in the tracked-changes version of the manuscript (shown in blue), and a detailed, point-by-point response is provided below.

Major Comments

1. Uncertainty and reproducibility of Raman-based quantification

The 3D Raman mapping and volumetric reconstruction form the technical foundation of this study. However, no explicit uncertainty propagation is provided for the calculated bulk compositions (Table 1). Please quantify and discuss the analytical uncertainties and reproducibility of the Raman-based measurements.

Reply: We thank the reviewer for this insightful comment. In the revised manuscript, we have taken several steps to further strengthen the robustness of our Raman analysis results.

(1) The three-dimensional (3D) Raman mapping, phase identification, and volumetric reconstruction approach used in this study follows the same analytical protocol established in our previous work (Jin et al., 2023), where the methodology was validated and shown to provide reliable quantitative results for multiphase fluid inclusions. We have now clarified this point explicitly in the revised manuscript (Lines 456–462).

(2) To directly assess analytical reproducibility, we performed independent repeat analyses on two representative fluid inclusions (No. 2 and No. 20). The results, reported in Tables S1 and S2, show highly consistent daughter mineral assemblages, phase volume proportions, and reconstructed bulk fluid compositions, demonstrating good reproducibility. This clarification has now been incorporated into the revised manuscript (Lines 476–480).

(3) We now report the standard deviations (SD) for the calculated contents of all major components (Table S2), providing a quantitative assessment of the analytical uncertainty associated with the Raman-based reconstruction.

(4) We have additionally included a ternary compositional diagram (Fig. S5) summarizing the reconstructed phase volume proportions and bulk fluid compositions of all 22 analyzed fluid inclusions. The tight clustering of data points in this diagram indicates limited compositional variability among inclusions and further supports the internal consistency and reproducibility

of the Raman reconstructed results.

Taken together, these multiple lines of evidence demonstrate that the Raman-based measurements are robust and reproducible, and that the reconstructed bulk compositions faithfully represent the original slab-derived fluids preserved within the inclusions.

Fig. S5 of the supplementary information. Phase proportions and reconstructed bulk compositions of multiphase fluid inclusions. (A) Ternary diagram showing relative phase proportions of the inclusions, with endmembers defined as brine (H₂O + dissolved NaCl), calcite, and chalcopyrite. (B) Ternary diagram illustrating reconstructed bulk compositions of the inclusions, with endmembers defined as H₂O + NaCl, CuFeS₂, and CaCO₃. In both diagrams, the data cluster tightly, indicating a high degree of compositional uniformity among the inclusions and supporting a common origin for the trapped fluids.

2. Variability among subduction settings

Please discuss potential variability of slab-derived sulfur contents among different subduction zones (e.g., Alps, Caledonides, Japanese arcs), and clarify the limitations of extrapolating the Sumdo data to global sulfur flux estimates.

Reply: We thank the reviewer for this important comment.

(1) We fully acknowledge that the precise sulfur contents of slab-derived fluids at sub-arc depths may vary among different subduction zones, such as the Alps, Caledonides, and Japanese arcs, reflecting differences in slab composition, thermal structure, redox state, and fluid–rock interaction histories. This potential variability is now explicitly stated in the revised

manuscript. Nevertheless, sulfur enrichment in sub-arc-depth fluids is not unique to the Sumdo system. Sulfur-rich daughter minerals and sulfur-bearing fluid inclusions have been widely reported from HP–UHP metamorphic rocks and associated veins in multiple subduction settings worldwide (Phillippot et al., 1991; Svensen et al., 1999, 2001; Frezzoti and Ferrando, 2015). In addition, geochemical studies of natural arc basalts consistently indicate sulfur enrichment in sub-arc mantle sources (de Hoog et al., 2001; Muth and Wallace, 2022; Taracsák et al., 2023), supporting an important role for slab-derived fluids in sulfur transfer. Importantly, the sulfur concentrations measured in this study (~3.2–10.3 wt.%) fall within the broader range predicted for subduction zone fluids from thermodynamic models and high-pressure experiments (Li et al., 2020; Jégo and Dasgupta, 2013, 2024; Beaudry and Sverjensky, 2024; He et al., 2024; Maffei et al., 2024). Our measurements not only validate previous experimental and theoretical predictions of elevated sulfur contents in sub-arc fluids but also provide more restricted natural constraints on the range of sulfur concentrations in slab-derived fluids. Collectively, these observations indicate that sulfur enrichment in slab-derived fluids at sub-arc depths is likely a common feature of deep subduction zones rather than an isolated characteristic of the Sumdo system. These clarifications are now explicitly stated in the revised manuscript (Lines 196–213).

(2) Furthermore, we have also clarified the assumptions, rationale, and limitations involved in extrapolating the Sumdo data to global sulfur budgets. (i) The Sumdo system records a cold subduction regime, which represents a widespread and geodynamically important class of subduction zones operating globally. Although results from a single system cannot capture the full range of global variability, cold subduction zones contribute substantially to global volatile recycling, allowing the extrapolating results from the Sumdo system to provide broadly applicable constraints on subduction zone sulfur budgets. (ii) The investigated samples record fluid compositions at sub-arc depths, representing a critical window of slab-derived sulfur release. As discussed above, sulfur-rich sub-arc fluids are not unique to the Sumdo system but have been documented in multiple subduction settings worldwide. Furthermore, our study not only validates previous experimental and theoretical predictions of elevated sulfur contents in sub-arc fluids but also provides more restricted natural constraints on the range of sulfur concentrations in slab-derived fluids. As such, we suggest that our measured sulfur concentrations can serve as an important and reasonable parameter for sulfur flux calculations. (iii) To further strengthen the calculation, we re-evaluated the second key parameter controlling sulfur fluxes, namely the global H₂O flux released from the altered oceanic crust. We performed thermodynamic calculations constrained by the geological and P–T conditions of the Sumdo system to estimate dehydration. The results indicate that approximately 60% of the water initially stored in altered oceanic crust is released at depths of 80–90 km. Based on this dehydration fraction, and integrating average H₂O contents of altered oceanic crust, global subduction zone length, and crustal thickness (Alt et al., 1999; Stern et al., 2020; Ribeiro et al., 2026), we estimate a global H₂O flux of ~82.5 Mt/y released from subducted oceanic crust at sub-arc depths. This estimate closely matches independent literature constraints (~91 Mt /y, Li et al., 2020), providing confidence in the adopted parameterization. (iv) Finally, based on these parameters, we calculate sulfur fluxes carried by sub-arc fluids and further estimate the efficiency of sulfur release from the slab at sub-arc depths, as well as its contribution to sulfur enrichment in arc systems. Importantly, these estimates are quantitatively consistent with

independent constraints derived from previous experimental, theoretical, and geochemical studies (Walters et al., 2020; Li et al., 2022; Taracsák et al., 2023), providing additional support for the reasonableness and reliability of extrapolating our results from the Sumdo system to global subduction zones.

Overall, we consider the extrapolation from the Sumdo system to broader assessments of sulfur cycling in subduction zones to be reasonable. We acknowledge that sulfur transfer fluxes may slightly vary among different subduction zones; however, our calculations provide an efficient and general assessment for understanding sulfur cycling in subduction zones at the global scale. These clarifications are now explicitly stated in the revised manuscript (Lines 331–360, 538–550).

3. Sulfide mineralogy and sulfur speciation

Chalcopyrite is the only sulfide phase observed in the fluid inclusions, whereas pyrite dominates in the host eclogite. Why is this the case? In addition, no oxidized S-bearing volatiles (e.g., SO_4^{2-}) were detected in the omphacite inclusions, although experimental and thermodynamic models suggest that oxidized sulfur species should be more mobile under subduction conditions. Please discuss this discrepancy.

Reply: We thank the reviewer for this constructive comment. In the revised manuscript, we have carefully re-evaluated sulfur speciation in the trapped fluids under UHP conditions and clarified the origin of the chalcopyrite within the fluid inclusion.

Based on the geological setting and P–T conditions of the Sumdo belt, we performed thermodynamic calculations using the Deep Earth Water (DEW) model. These calculations indicate that sulfur in slab-derived fluids at sub-arc depths is dominantly present as oxidized species, primarily SO_4^{2-} and HSO_4^- (Fig. S7). This result suggests that sulfur was initially entrapped in the fluid inclusions predominantly in an oxidized form, consistent with previous experimental and thermodynamic studies predicting high mobility of oxidized sulfur species under deep subduction conditions (Walters et al., 2020; Beaudry and Sverjensky, 2024; Maffei et al., 2024).

We further propose that the sulfur speciation recorded in the fluid inclusions may have been modified after entrapment. Electron probe microanalysis (EPMA) of host omphacite across fluid inclusions exposed at the surface reveals slightly but systematically lower Fe contents in omphacite adjacent to inclusions compared with regions farther away (Fig. S6). Although the magnitude of this variation is small, it suggests that post-entrapment Fe exchange between the trapped fluids and the host omphacite may have promoted localized redox reactions, leading to partial reduction of sulfur species and subsequent precipitation of chalcopyrite within the inclusions. Alternatively, hydrogen diffusion across the inclusion–host boundary may also have contributed to post-entrapment redox modification, although this process is difficult to quantify directly.

Taken together, we now clarify in the revised manuscript (Lines 226–253) that sulfur in slab-derived fluids at sub-arc depths was initially present predominantly in oxidized forms, whereas post-entrapment interactions between the trapped fluids and the host omphacite likely resulted in sulfur reduction and the precipitation of chalcopyrite as the dominant sulfide phase in the inclusions.

Fig. S6 of the supplementary information. Major-element variations in host omphacite across a representative multiphase fluid inclusion. As shown in (A), the host omphacite exhibits slight but systematic variations (0.4 wt.%) in Fe content, with lower Fe concentrations observed in regions adjacent to the fluid inclusions compared to areas farther away. This Fe variation is consistent with limited Fe exchange between the trapped fluids and the host omphacite, which may have facilitated post-entrapment redox modification of sulfur species within the fluid inclusions.

Fig. S7 of the supplementary information. Thermodynamic modeling of sulfur speciation in subduction zone fluids constrained by the Sumdo system. The results show that slab-derived

fluids at sub-arc depths are dominated by oxidized sulfur species (e.g., SO_4^{2-} and HSO_4^-).

4. Role of calcite and carbonate components

What is the role of calcite in the fluid inclusions? Could CO_3^{2-} enhance sulfide solubility instead of (or in addition to) NaCl? This should be addressed quantitatively or at least conceptually.

Reply: We thank the reviewer for this insightful comment. The presence of calcite within the inclusions indicates that the trapped fluids were carbon-enriched at the time of entrapment, likely with CO_2 , CO_3^- , or HCO_3^- as an important dissolved carbon component (Fig. S7).

With respect to whether CO_3^{2-} directly enhances sulfide solubility, we note that, to our knowledge, no experimental studies have quantitatively evaluated the effect of CO_3^{2-} on sulfur solubility in deep subduction zone fluids. However, observations from natural samples consistently show that slab-derived supercritical fluids recorded in fluid inclusions for UHP rocks are commonly enriched in both sulfur and carbon (e.g., Zhang et al., 2008; Jin et al., 2023), suggesting a close association between carbon-rich fluids and sulfur enrichment.

Recent studies suggest that the presence of carbon can lower the P–T conditions of the second critical endpoint in supercritical fluids (Li et al., 2020; Ni et al., 2025). Supercritical fluids generated under such conditions are characterized by enhanced solubilities for volatiles (e.g., sulfur) and metal elements. We therefore suggest that carbon plays an indirect role in promoting sulfur enrichment in slab-derived fluids. The estimated trapping conditions of the Sumdo inclusions (~2.7 GPa, Yang et al., 2009) approach the second critical endpoint of supercritical fluids reported for mafic systems at ~3.0 GPa (Mibe et al., 2011), and the reconstructed fluid compositions exhibit characteristics consistent with near-supercritical or supercritical behavior. We therefore suggest that the Sumdo inclusions most likely record supercritical fluids, in which sulfur enrichment is facilitated by the physicochemical properties of the supercritical state. In addition, following your insightful suggestion, we now describe the entrapped fluids in the multiphase inclusions as near-supercritical, solute-rich fluids.

Moreover, guided by recent thermodynamic results (Beaudry and Sverjensky, 2024), we highlight that Ca and Na cations in fluids can substantially enhance sulfate solubility under subduction zone conditions. Beaudry and Sverjensky (2024) demonstrate that oxidized sulfur can be efficiently mobilized in high-pressure fluids through complexing with Ca and Na (e.g., CaHSO_4^+ and Na_2SO_4^0), resulting in significantly enhanced sulfur solubility under sub-arc conditions. This interpretation is fully consistent with our observations: the Sumdo fluid inclusions are enriched in both CaO (10.16 wt.%) and Na (3.22 wt.%), and the measured sulfur concentrations (~3.2–10.3 wt.%) agree well with experimentally and thermodynamically predicted solubilities at comparable P–T conditions.

Collectively, we therefore interpret Ca and Na as the primary controls on sulfur solubility in these fluids, while carbonate components exert an indirect influence by modifying fluid physicochemical properties. These interpretations have now been clarified in the revised manuscript (Lines 263–272, 288–293, 308–314).

5. Potential overestimation of global sulfur fluxes

The limited omphacite inclusions from UHP eclogites in the Sumdo belt may represent locally sulfur-enriched portions of the subducting slab. Consequently, using these inclusions to estimate global slab-derived sulfur contributions might lead to overestimation. Please discuss

this limitation explicitly.

Reply: We thank the reviewer for raising this constructive comment. In the revised manuscript, we now explicitly discuss and clarify the limitations, assumptions, and rationale underlying our sulfur flux estimates.

(1) Representativeness and limitations of the Sumdo system. The Sumdo belt represents a well-characterized cold subduction zone system, which constitutes a dominant end-member of global subduction styles, despite the presence of less common hot subduction zones. We therefore argue that the Sumdo system provides a reasonable analogue for global cold subduction environments.

(2) Consistency with sulfur-rich fluids at sub-arc depths. The sulfur concentrations measured directly in the Sumdo fluid inclusions are consistent with recent experimental and thermodynamic predictions of sulfur-rich slab-derived fluids at sub-arc depths (e.g., Li et al., 2020; Jégo and Dasgupta, 2013, 2024; Beaudry and Sverjensky, 2024; He et al., 2024; Maffei et al., 2024). These independent constraints suggest that sulfur enrichment in slab-derived fluids at sub-arc depths is likely a recurrent feature of deep subduction systems, rather than an isolated characteristic unique to Sumdo.

(3) Water flux constraints. Because H₂O flux is a key parameter in sulfur flux calculations, we now include a dehydration model constrained by the geological and P–T conditions of the Sumdo system (2.6–2.7 GPa and 630–780 °C, Yang et al., 2009). These calculations indicate that ~60% of the water initially stored in altered oceanic crust is released at sub-arc depths. Scaling this result to the global subduction system using published average H₂O contents of altered oceanic crust (Alt et al., 1999), a global subduction zone length of ~55,000 km (Stern et al., 2002), and an altered crustal thickness of ~0.6 km (Ribeiro et al., 2026) yields an estimated H₂O release flux of ~84.5 Mt/yr. This value closely matches independent global estimates of ~91 Mt/yr at ~90 km depth (Li et al., 2020), supporting the robustness of our water-flux parameterization.

(4) Sulfur flux estimates and uncertainty. Using directly measured sulfur concentrations in slab-derived fluids together with reasonably calculated H₂O fluxes, we calculate sulfur fluxes at sub-arc depths and compare them with both the sulfur input from subducted altered oceanic crust and sulfur outputs from arc systems. We now explicitly incorporate uncertainty by accounting for the observed variability (standard deviation) in sulfur concentrations, yielding an estimated contribution of $70 \pm 20\%$ of arc sulfur output from slab-derived fluids. This range overlaps with previous indirect estimates (e.g., Walters et al., 2020; Li et al., 2022; Taracsák et al., 2023), suggesting that our results do not represent a substantial overestimation.

Collectively, we acknowledge that sulfur transfer fluxes may slightly vary among different subduction zones. However, as we discussed in detail above, we consider the extrapolation from the Sumdo system to broader assessments of sulfur cycling in subduction zones to be reasonable. These clarifications are now explicitly stated in the revised manuscript (Lines 331–360, 538–558). We sincerely hope this clarification can meet the reviewer's approval!

Minor Comments

Line 118: Please include full microscopic images of the mineral assemblage in the Supplementary Materials and specify the trapping stage of the fluids (during subduction or exhumation). Are the omphacites vein minerals, as reported in other subduction settings?

Reply: We thank the reviewer for this helpful comment. Representative microscopic images documenting the mineral assemblages of the Sumdo eclogites have now been added to the Supplementary Information (Fig. S2). The investigated eclogites are composed predominantly of garnet and omphacite, with minor amphibole, rutile, and pyrite (Fig. S2).

Fig. S2 in the Supplementary information. Representative photomicrographs illustrating the mineral assemblages of the Sumdo eclogites. The samples are composed predominantly of omphacite (Omp) and garnet (Grt), with minor rutile (Rt), amphibole (Amp), and pyrite (Py).

Furthermore, previous petrological and geochemical evidence indicates that omphacite in the Sumdo eclogites formed during a single metamorphic stage, most likely corresponding to peak UHP conditions (Liu et al., 2019b). The omphacite compositions are generally homogeneous, with jadeite contents ranging from ~20 to 39 mol.% (Liu et al., 2019b). Accordingly, the fluid inclusions hosted in omphacite are interpreted to have been entrapped at or near peak metamorphic conditions, thereby providing reliable records of slab-derived fluids at sub-arc depths under UHP conditions.

Generally, omphacite is a characteristic eclogite-facies mineral and occurs as elongated prismatic crystals in the Sumdo eclogites. Following the principle of surface-energy minimization during inclusion entrapment, primary fluid inclusions commonly adopt morphologies that reflect the crystallographic habit of their host minerals. The prismatic morphology of the inclusions hosted in omphacite is therefore interpreted as a key textural feature indicative of primary fluid entrapment, rather than secondary trapping during exhumation. Similar prismatic fluid inclusions hosted in omphacite have been widely documented in HP–UHP metamorphic rocks and associated veins from subduction zones worldwide (e.g., Zhang et al., 2008; Frezzotti and Ferrando, 2015; Jin et al., 2023). These clarifications have now been incorporated into the revised manuscript (Lines 130–135, 139–142, 209–211).

Lines 144–145: Clarify why chalcopyrite appears as the only sulfide phase within the inclusions, while pyrite dominates the host eclogite. Does chalcopyrite have higher solubility under subduction conditions?

Reply: We thank the reviewer for this insightful comment. As discussed above and now clarified in the revised manuscript (Lines 226–253), we provide a detailed interpretation for the origin of chalcopyrite within the fluid inclusions. We infer that sulfur in the fluids was dominantly present as oxidized species at the time of entrapment. We infer that sulfur in the fluids was dominantly present as oxidized species at the time of entrapment. Following entrapment, post-entrapment redox modification—most likely driven by limited Fe exchange between the host omphacite and the trapped fluids—may have led to partial reduction of sulfur within the inclusions. Under these conditions, reduced sulfur species could have reacted with dissolved Cu and Fe in the co-entrapped fluids, resulting in chalcopyrite precipitation. We therefore interpret the occurrence of chalcopyrite within the inclusions not as evidence for intrinsically higher chalcopyrite solubility under subduction zone conditions, but rather as the product of post-entrapment redox reactions and metal availability within a closed inclusion system.

Line 146: The pronoun “that” is ambiguous—does it refer to pyrite or chalcopyrite? Please clarify whether decomposition is pressure- or temperature-driven, and cite any reports of pyrite in metamorphic fluid inclusions.

Reply: We apologize for the ambiguity caused by our previous wording. In the original manuscript, our intention was to distinguish between sulfide phases occurring in the host rock and those present within the fluid inclusions: sulfides in the host eclogite are dominantly pyrite, whereas chalcopyrite is the only sulfide observed in the fluid inclusions. This distinction indicates that the sulfides in the inclusions are unlikely to result from mechanical entrainment or physical transfer from the host minerals.

In the revised manuscript, we clarify that sulfur was most likely initially trapped in the fluid inclusions as oxidized species. In addition, recent experimental and thermodynamic studies further demonstrate that pyrite can act as a source of oxidized sulfur in slab-derived fluids under subduction zone conditions (2–3 GPa, 400–800 °C) across a broad range of oxygen fugacities (e.g., Beaudry & Sverjensky, 2024). This mechanism is consistent with observations from the Sumdo eclogites, including the presence of residual pyrite in the host rocks (Fig. S2) and the close correspondence between sulfur-rich fluid entrapment conditions (~2.6–2.7 GPa, 630–780 °C; Yang et al., 2009) and those predicted for oxidized sulfur mobilization. We have rephrased and reorganized the relevant discussion accordingly in the revised manuscript (Lines 226–253).

Line 176: Explain why reduced sulfur species (S^{2-}) are released rather than oxidized ones (SO_3^{2-}). Note that reduced sulfur may be re-trapped by Fe^{2+} in the host rocks.

Reply: We thank the reviewer for this insightful comment. In the revised manuscript, we clarify that sulfur in the fluids initially trapped by the inclusions was dominantly present as oxidized species rather than reduced S^{2-} . This interpretation is supported by thermodynamic modelling under the P–T conditions of the Sumdo system (Fig. S7). We further propose that post-entrapment fluid–rock interaction, particularly Fe^{2+} exchange between the trapped fluids and host omphacite, may have facilitated partial redox re-equilibration. Such localized redox modification could have promoted the reduction of initially oxidized sulfur species within the inclusions, leading to the generation of reduced sulfur species and the precipitation of sulfide phases. This revised interpretation has now been clarified in the revised manuscript (Lines 226–

253).

Lines 221–222, 235–236: Provide details about the salinity measurements and indicate whether NaCl is present in the Sumdo inclusions. Could CO_3^{2-} enhance sulfide solubility?

Reply: We thank the reviewer for these insightful suggestions. Details of the salinity measurements were previously reported by Liu et al. (2019b), who documented fluid inclusions spanning a wide range of salinities. The inclusions investigated in the present study specifically correspond to fluids trapped at peak metamorphic conditions at sub-arc depths, and are characterized by high salinities of approximately 10–22 wt.%. For the purpose of bulk composition reconstruction, we adopted an average salinity of 15 wt.%. In addition, no solid NaCl daughter minerals were observed in the Sumdo fluid inclusions, suggesting that NaCl was predominantly present as a dissolved component in the aqueous phase. These points have now been explicitly clarified in the Methods section (Lines 451–453, 467–471).

With respect to the potential role of CO_3^{2-} in enhancing sulfide solubility, we agree that this is an important yet currently unresolved issue. To our knowledge, no experimental studies have directly quantified the influence of carbon on sulfur solubility in UHP fluids, and we therefore avoid speculative interpretations. Nevertheless, as discussed above, we have added a concise discussion in the revised manuscript proposing a possible indirect role of carbon in sulfur enrichment in subduction zone fluids. Specifically, carbon may lower the pressure conditions required for the generation of supercritical fluids (Li et al., 2020; Ni et al., 2025), thereby modifying fluid properties and potentially enhancing sulfur solubility indirectly. These clarifications and additions have now been incorporated into the revised manuscript (Lines 263–272).

Line 234: The discussion of “supercritical-like” fluids is speculative without supporting evidence. Use this term cautiously or replace it with “near-critical, solute-rich fluids,” and cite relevant experimental studies (e.g., Mibe et al., 2011).

Reply: We thank the reviewer for this helpful suggestion and agree that the term “supercritical-like” should be used with caution. In our study, the reconstructed compositions of the multiphase fluid inclusions are consistent with those expected for supercritical fluids, and the estimated pressure conditions of fluid entrapment (~2.7Gpa, Yang et al., 2009) approach the second critical endpoint reported for mafic systems in experimental studies (3.0 Gpa, Mibe et al., 2011). This combination of chemical characteristics and near-critical pressure conditions motivated our original use of the term. However, to avoid over-interpretation and to adopt more rigorous terminology, we have revised the manuscript to replace “supercritical-like fluids” with “near-supercritical, solute-rich fluids” throughout the text (Lines 256, 269, 272, 311). We have also added appropriate experimental references to support this revised interpretation.

Lines 248–250: When calculating the slab-derived sulfur flux, it would be helpful to provide the estimated water flux for the Sumdo subduction zone, rather than relying solely on global averages.

Reply: We thank the reviewer for this constructive comment. In the revised manuscript, we now provide an explicit estimate of the slab-derived water flux. We performed thermodynamic modelling constrained by the geological and P–T conditions of the Sumdo system to quantify

dehydration of altered oceanic crust. The results indicate that approximately 60% of the water initially stored in the altered oceanic crust is released at sub-arc depths (Fig. S7). Given the similarity between the Sumdo system and other cold subduction regimes worldwide, we treat Sumdo as representative of altered oceanic crust dehydration in cold subduction settings. We then scale this dehydration estimate to the global subduction system using previously reported average H₂O contents of altered oceanic crust (Alt et al., 1999), a global subduction zone length of ~55,000 km (Stern et al., 2002), and an altered crustal thickness of ~0.6 km (Ribeiro et al., 2026). This approach yields an estimated H₂O release flux of ~84.5 Mt/yr at 80–90 km of sub-arc depths. This value is consistent with independent global assessments, which suggest a slab-derived water flux of ~91 Mt/yr at ~90 km depth (Li et al., 2020). The close agreement between these estimates supports the robustness of our water-flux parameterization. We therefore consider the revised H₂O flux (~84.5 Mt/yr) adopted in our subsequent sulfur flux calculations to be reasonable and well constrained. These clarifications and calculations have now been incorporated into the revised manuscript (Lines 505–534, 538–550).

Line 295: Include details of laser power and acquisition parameters in the Methods section to ensure reproducibility.

Reply: We thank the reviewer for this helpful suggestion. In the revised manuscript (Lines 430–434), we have now explicitly clarified the key Raman analytical parameters to ensure reproducibility.

For the reviewer’s convenience, the newly added text reads: “The laser power at the sample surface was maintained at ~25 nW, which is sufficiently low to minimize laser-induced heating or damage to the inclusions while still ensuring robust detection of Raman signals from both the daughter minerals and associated fluid phase. Spectra were collected over the range of 100–4,400 cm⁻¹ to fully capture the Raman features of the multiphase assemblages.”

Lines 328–329: The global sulfur input flux from subducted altered oceanic crust (~48.09 Mt/yr) may not directly apply to the Sumdo subduction zone. Please provide uncertainty ranges (e.g., 75 ± ? %) to reflect the robustness of your global estimate.

Reply: We thank the reviewer for this insightful comment. In the revised manuscript, we clarify that the global sulfur input flux from subducted altered oceanic crust (~48.09 Mt/yr, Evans, 2012) is used as a reference framework, rather than being directly applied to the Sumdo subduction zone. Because the Sumdo system represents a cold subduction end-member—corresponding to a widespread and geodynamically significant class of global subduction regimes—it provides a reasonable basis for cautious extrapolation to broader subduction zone contexts.

To strengthen this extrapolation, we incorporate thermodynamic dehydration modeling based on the P–T conditions of the Sumdo system and apply it to estimate the global H₂O release flux from the altered oceanic crust at sub-arc depths (~80–90 km). The resulting H₂O flux (~84.5 Mt/yr) closely matches previously published global estimates (~91 Mt/yr, Li et al., 2020) cited in the original manuscript, providing confidence in the water-flux parameter used. Moreover, the sulfur concentrations measured in this study (~3.2–10.3 wt.%) fall within the broader range inferred for subduction zone fluids from thermodynamic models and high-pressure experiments (~0.5–15 wt.%) (Li et al., 2020, Jégo and Dasgupta, 2013, 2014, Beaudry and Sverjensky, 2024;

He et al., 2024; Maffei et al., 2024), but define a more restricted range. Together, on this basis, we calculate sulfur fluxes carried by sub-arc fluids derived from the subducted altered oceanic crust and compare them with the total sulfur input from subducted altered oceanic crust to provide a general assessment of sulfur release efficiency at sub-arc depths and its potential contribution to arc sulfur output. Importantly, we now explicitly include uncertainty associated with sulfur concentrations by incorporating the measured standard deviation (SD) of sulfur contents in the fluid inclusions. Using an average sulfur concentration of 6.3 ± 1.8 wt.%, our revised calculations indicate that slab-derived fluids may contribute $70 \pm 20\%$ of the sulfur released to arc magmatic systems. This uncertainty range overlaps well with independent estimates derived from natural arc rocks and melt inclusions (40–70%, Taracsák et al., 2023), supporting the robustness of our calculations.

These clarifications have been incorporated into the revised manuscript (Lines 331–361, 381–385). We hope that this revised explanation satisfactorily addresses the reviewer's concern.

We once again sincerely thank the reviewer for recognizing the novelty of our work and for the constructive and insightful suggestions. These comments have significantly improved the clarity, rigor, and overall quality of the manuscript!

References

- Alt, J. C. & Teagle, D. A. H. The uptake of carbon during alteration of ocean crust. *Geochim. Cosmochim. Acta* 63, 1527–1535 (1999).
- Beaudry, P. & Sverjensky, D. A. Oxidized sulfur species in slab fluids as a source of enriched sulfur isotope signatures in arcs. *Geochem. Geophys. Geosyst.* 25, e2024GC011542 (2024).
- de Hoog, J. C. M., Taylor, B. E. & van Bergen, M. J. Sulfur isotope systematics of basaltic lavas from Indonesia: implications for the sulfur cycle in subduction zones. *Earth Planet. Sci. Lett.* 189, 237–252 (2001).
- de Moor, J. M., Fischer, T. P. & Plank, T. Constraints on the sulfur subduction cycle in Central America from sulfur isotope compositions of volcanic gases. *Chem. Geol.* 588, 120627 (2022).
- Evans, K. A. The redox budget of subduction zones. *Earth-Sci. Rev.* 113, 11–32 (2012).
- Farsang, S. & Zajacz, Z. Sulfur species and gold transport in arc magmatic fluids. *Nat. Geosci.* 18, 98–104 (2025).
- Frezzotti, M. L. & Ferrando, S. The chemical behavior of fluids released during deep subduction based on fluid inclusions. *Am. Mineral.* 100, 352–377 (2015).
- Goldstein, R. H. Fluid inclusions in sedimentary and diagenetic systems. *Lithos* 55, 159–193 (2001).
- He, D. Y., Qiu, K. F., Simon, A. C., Pokrovski, G. S., Yu, H. C., Connolly, J. A. D., Li, S. S., Turner, S., Wang, Q. F., Yang, M. F. & Deng, J. Mantle oxidation by sulfur drives the formation of giant gold deposits in subduction zones. *Proc. Natl Acad. Sci. U.S.A.* 121, e2404731121 (2024).
- Hogg, O. R., Edmonds, M., Wieser, P. E., Gleeson, M., Jenner, F. E. & Blundy, J. Copper-rich fluids arising from sulfide resorption by hydrous arc melts. *Sci. Rep.* 16, 29115 (2026).
- Jenner, F. E. Cumulate causes for the low contents of sulfide-loving elements in the continental crust. *Nat. Geosci.* 10, 524–528 (2017).
- Jégo, S. & Dasgupta, R. Fluid-present melting of sulfide-bearing ocean-crust: Experimental constraints on the transport of sulfur from subducting slab to mantle wedge. *Geochim. Cosmochim. Acta* 110, 106–134 (2013).
- Jégo, S. & Dasgupta, R. The fate of sulfur during fluid-present melting of subducting basaltic crust at variable oxygen fugacity. *J. Petrol.* 55, 1019–1050 (2014).
- Jin, D., Xiao, Y., Tan, D. B., Wang, Y. Y., Wang, X., Li, W., Su, W. & Li, X. Supercritical fluid in deep subduction zones as revealed by multiphase fluid inclusions in an ultrahigh-pressure metamorphic vein. *Proc. Natl Acad. Sci. U.S.A.* 120, e2219083120 (2023).
- Kepezhinskas, P., Defant, M. J. & Widom, E. Abundance and distribution of PGE and Au in the island-arc mantle: implications for sub-arc metasomatism. *Lithos* 60, 113–128 (2002).
- Lee, C. T., Luffi, P., Chin, E. J., Bouchet, R., Dasgupta, R., Morton, D. M., Le Roux, V., Yin, Q. Z. & Jin, D. Copper systematics in arc magmas and implications for crust–mantle differentiation. *Science* 336, 64–68 (2012).
- Lee, C.-T. A. & Tang, M. How to make porphyry copper deposits. *Earth Planet. Sci. Lett.* 529, 115868 (2020).
- Li, J.-L., Gao, J., John, T., Klemd, R. & Su, W. Fluid-mediated metal transport in subduction zones and its link to arc-related giant ore deposits: constraints from a sulfide-bearing HP

- vein in lawsonite eclogite (Tianshan, China). *Geochim. Cosmochim. Acta* 120, 326–362 (2013).
- Li, J. L., Schwarzenbach, E. M., John, T., Ague, J. J., Huang, F., Gao, J., Klemd, R., Whitehouse, M. J. & Wang, X. S. Uncovering and quantifying the subduction zone sulfur cycle from the slab perspective. *Nat. Commun.* 11, 514 (2020).
- Li, J.-L., Schwarzenbach, E. M., John, T., Ague, J. J., Tassara, S., Gao, J. & Konecke, B. A. Subduction zone sulfur mobilization and redistribution by intraslab fluid–rock interaction. *Geochim. Cosmochim. Acta* 297, 40–64 (2021).
- Li, W. & Ni, H. Dehydration at subduction zones and the geochemistry of slab fluids. *Sci. China Earth Sci.* 63, 1925–1937 (2020).
- Li, Y.-B., Chen, Y., Su, B., Zhang, Q.-H. & Shi, K.-H. Redox species and oxygen fugacity of slab-derived fluids: Implications for mantle oxidation and deep carbon–sulfur cycling. *Front. Earth Sci.* 10, (2022).
- Liu, H., Sun, H., Xiao, Y., Wang, Y., Zeng, L., Li, W., Guo, H., Yu, H. & Pack, A. Lithium isotope systematics of the Sumdo eclogite, Tibet: tracing fluid–rock interaction of subducted low-Taltered oceanic crust. *Geochim. Cosmochim. Acta* 246, 385–405 (2019a).
- Liu, H., Xiao, Y., van den Kerkhof, A., Wang, Y., Zeng, L. & Guo, H. Metamorphism and fluid evolution of the Sumdo eclogite, Tibet: constraints from mineral chemistry, fluid inclusions and oxygen isotopes. *J. Asian Earth Sci.* 172, 292–307 (2019b).
- Lorand, J.-P., Luguet, A. & Alard, O. Platinum-group element systematics and petrogenetic processing of the continental upper mantle: a review. *Lithos* 164–167, 2–21 (2013).
- Maffei, A., Frezzotti, M. L., Connolly, J. A. D., Castelli, D. & Ferrando, S. Sulfur disproportionation in deep COHS slab fluids drives mantle wedge oxidation. *Sci. Adv.* 10, adj2770 (2024).
- McInnes, B. I. A., McBride, J. S., Evans, N. J., Lambert, D. D. & Andrew, A. S. Osmium isotope constraints on ore metal recycling in subduction zones. *Science* 286, 512–516 (1999).
- Métrich, N., Schiano, P., Clocchiatti, R. & Maury, R. C. Transfer of sulfur in subduction settings: an example from Batan Island (Luzon volcanic arc, Philippines). *Earth Planet. Sci. Lett.* 167, 1–14 (1999).
- Mibe, K., Kawamoto, T., Matsukage, K. N., Fei, Y. & Ono, S. Slab melting versus slab dehydration in subduction-zone magmatism. *Proc. Natl Acad. Sci. U.S.A.* 108, 8177–8182 (2011).
- Muth, M. J. & Wallace, P. J. Sulfur recycling in subduction zones and the oxygen fugacity of mafic arc magmas. *Earth Planet. Sci. Lett.* 599, 117833 (2022).
- Newton, R. C. Solubility of anhydrite, CaSO₄, in NaCl – H₂O solutions at high pressures and temperatures: Applications to fluid – rock interaction. *J. Petrol.* 46, 701 – 716 (2004).
- Ni, H., Xiao, Y., Xiong, X., Liu, X., Gao, C., Chen, Y.-X., Li, Y., Li, W.-C., Guo, X., Wang, Y.-Y., Tan, D.-B. & Zhang, L. Formation and evolution of supercritical geofluid. *Sci. China Earth Sci.* 68, 39–51 (2025).
- Philippot, P. & Selverstone, J. Trace-element-rich brines in eclogitic veins: Implications for fluid compositions and transport during subduction. *Contrib. Mineral. Petrol.* 106, 417–430 (1991).

- Ren, A., Wang, Z., Aulbach, S., Zong, K., Wang, X., Zou, Z., Shen, Y., Cheng, H., Hu, Z. & Zhu, Z. Subduction-related transfer of sulfur and chalcophile elements recorded in continental mantle wedge peridotites. *Geochim. Cosmochim. Acta* 398, 11–28 (2025).
- Ribeiro, J. M., Lin, J. & Ryan, J. The serpentinized oceanic mantle: a potentially substantial volatile sink. *Earth Planet. Sci. Lett.* 674, 119730 (2026).
- Richards, J. P. The oxidation state, and sulfur and Cu contents of arc magmas: implications for metallogeny. *Lithos* 233, 27–45 (2015).
- Stern, R. J. Subduction zones. *Rev. Geophys.* 40, 1012 (2002).
- Svensen, H., Jamtveit, B., Yardley, B., Engvik, A. K., Austrheim, H. & Broman, C. Lead and bromine enrichment in eclogite-facies fluids: extreme fractionation during lower-crustal hydration. *Geology* 27, 467–470 (1999).
- Svensen, H., Jamtveit, B. & Banks, D. A. Halogen contents of eclogite facies fluid inclusions and minerals: Caledonides, western Norway. *J. Metamorph. Geol.* 19, 165–178 (2001).
- Tan, D. B., Lei, J., Xiao, Y., Gu, H. O., Sun, H., Ye, X., & Guo, Z. L. Carbonic Fluids Drive Continental Carbon Cycling as Revealed by the Geochemistry of the Eclogite-Garnet Peridotite Interface. *Geophysical Research Letters*, 52(24). (2025).
- Taracsák, Z., Mather, T. A., Ding, S., Plank, T., Brounce, M., Pyle, D. M. & Aiuppa, A. Sulfur from the subducted slab dominates the sulfur budget of the mantle wedge under volcanic arcs. *Earth Planet. Sci. Lett.* 602, 117948 (2023).
- Taracsák, Z., Hartley, M. E., Burgess, R., Edmonds, M., Longpré, M. A., Monteleone, B. D., Tartèse, R. & Turchyn, A. V. The origin of sulfur in Canary Island magmas and its implications for Earth's deep sulfur cycle. *Proc. Natl Acad. Sci. U.S.A.* 122, e2416070122 (2025).
- Walters, J. B., Cruz-Uribe, A. M. & Marschall, H. R. Sulfur loss from subducted altered oceanic crust and implications for mantle oxidation. *Geochem. Perspect. Lett.* 13, 36–41 (2020).
- Walters, J. B., Cruz-Uribe, A. M., Marschall, H. R. & Boucher, B. The role of sulfides in the chalcophile and siderophile element budget of the subducted oceanic crust. *Geochim. Cosmochim. Acta* 304, 191–215 (2021).
- Weller, O. M., St-Onge, M. R., Rayner, N., Waters, D. J., Searle, M. P. & Palin, R. M. U–Pb zircon geochronology and phase equilibria modelling of a mafic eclogite from the Sumdo complex of south-east Tibet: insights into prograde zircon growth and the assembly of the Tibetan Plateau. *Lithos* 262, 729–741 (2016).
- Yang, J., Xu, Z., Li, Z., Xu, X., Li, T., Ren, Y., Li, H., Chen, S. & Robinson, P. T. Discovery of an eclogite belt in the Lhasa block, Tibet: a new border for Paleo-Tethys? *J. Asian Earth Sci.* 34, 76–89 (2009).
- Zhang, Z.-M., Shen, K., Sun, W.-D., Liu, Y.-S., Liou, J. G., Shi, C. & Wang, J.-L. Fluids in deeply subducted continental crust: petrology, mineral chemistry and fluid inclusion of UHP metamorphic veins from the Sulu orogen, eastern China. *Geochimica et Cosmochimica Acta* 72, 3200–3228 (2008).
- Zhao, S.-Y., Yang, A. Y., Langmuir, C. H. & Zhao, T.-P. Oxidized primary arc magmas: constraints from Cu/Zr systematics in global arc volcanics. *Sci. Adv.* 8, eabk0718 (2022).
- Zhu, J., Zhang, L., Li, H., Jiang, R., Zhang, L. & Tao, R. The effect of iron sulfidation on the stability of the trisulfur radical ion $S_3^{\bullet-}$ in subduction zone fluids. *Geochim. Cosmochim. Acta* 415, 40 – 55 (2026).

Point-by-point response

Reviewer 1 Dr. Zoltán Taracsák

I read the revised version of the ms. You sufficiently addressed the major comments, and I think the manuscript in its present form is much more convincing and suitable for publication in Nature Communications. I appreciate the extra modelling work and EPMA analyses you carried out. I have some minor comments and suggestions to the text where I think you should be more nuanced as I indicate below (these should be simple to do by adding a sentence or two)– once these are sorted, I recommend publication.

Reply: We sincerely thank Dr. Zoltán Taracsák for the positive evaluation of our manuscript and for the insightful suggestions that have helped further improve its clarity and robustness. We have carefully considered all the comments raised in this round, and the corresponding revisions have been incorporated into the manuscript with tracked changes (shown in blue). Detailed point-by-point responses are provided below.

Line62: Maybe reword the start of the sentence starting with “more fundamentally” – I suggest “It is also possible such discrepancy arises”...

Reply: Ok, the sentence has been revised accordingly (Lines 54-56) and now reads: “However, it may also arise from the lack of direct constraints on sulfur concentrations and fluxes in natural slab-derived fluids themselves.”

Line125-142: It is good that you added this segment - but I still have an issue around the use of "primary origin"-which strictly means that the FIs were entrapped during growth of the omphacite cores. Crosscutting inclusions with homogeneous fill may still be secondary relative to the core, placing FI formation between growth of omphacite cores and rims. I accept the arguments here that these FIs represent near-peak P-T conditions (particularly as these are in the core only), and representative of fluids lost from slabs-but you can say that without confirming primary origin (as you do in lines 139-142), or by mentioning that even if the FIs formed after the omphacite cores crystallised, they predate the rims, which is firm evidence of eclogite facies origin.

Reply: We thank the reviewer for this important suggestion. We have removed the expression “primary origin” from the manuscript (Lines 119-134). Instead, we now describe the inclusions more cautiously by emphasizing that they occur exclusively within the cores of omphacite. Because omphacite is a high-pressure mineral stable under eclogite-facies conditions, the occurrence of fluid inclusions within its cores indicates that these inclusions record fluids present at or near peak metamorphic conditions. As such, they provide critical constraints on the composition of slab-derived fluids at sub-arc depths. These revisions improve the precision of the description while retaining the main conclusions of the study.

Line164: again, use of primary fluid inclusion. I would leave primary out, and just state that these fluids were present concurrent to omphacite crystallisation at high P-T.

Reply: Ok, the sentence has been revised accordingly (Lines 155-156) and now reads: “First, as discussed above, the investigated inclusions were trapped during omphacite crystallization at peak metamorphic conditions.”

Line181-184: I agree that post-entrapment addition/loss of sulfur and Cu can be ruled out - but for elements like Ca and Fe, the contribution from the host may still be non-negligible. I appreciate that you carried out the EPMA work, but an issue that remains is that the silicate-host may have undergone diffusive re-equilibration too (partially or fully). I recommend changing the text here slightly to add nuance (i.e. add that modification was negligible for element other than Ca and Fe).

Reply: We thank the reviewer for pointing this out. For clarity and rigor, we have revised the relevant sentence in the manuscript (Lines 172-174). The revised text now reads: “This pattern suggests localized post-entrapment element exchange. However, the overall extent of modification appears negligible for elements other than Ca and Fe and is unlikely to have substantially altered the bulk composition of the trapped fluids.”

Line198-202 I don't think these two sentences add to your argument, considering that the quoted experiment/model based S range is, while broader, it still fully overlaps with your FIs. I would delete these two sentences and only leave in the revised text.

Reply: Thanks. As recommended, we have removed these two sentences from the manuscript.

Line276: delete squarely

Reply: Ok, we have deleted it.

Line278: mention where Dabie Shan is for broader context, and what subduction system it represent.

Reply: We have added a brief description of the Dabie Shan in the revised manuscript (Lines 264-265), noting that it (Dabie orogenic belt) represents one of the well-known continental subduction zones in China.

Line290: Maybe this is one of my last real concern with the ms - I think it would be important to highlight here that Ca contents may be affected by exchange with the host here, and also the fact that you did not measure Na in the brine, but assumed a 15% salinity based on a published dataset that had a range of 12-22%. In line 291 you state these concentrations are observed, but really Na is just estimated or implied. You should point out these limitations at the end of the paragraph.

Reply: We thank the reviewer for this important comment. We agree that the Ca concentrations in the fluid inclusions may potentially be influenced by exchange with the host mineral. Indeed, our EPMA elemental mapping shows a variation of up to ~0.6 wt.% in Ca, suggesting that limited modification may occur. However, such modification is unlikely to significantly alter the overall Ca-rich nature of the fluids. Ca-rich fluids are commonly observed in subduction environments, and in this case, the high carbon contents observed in the inclusions likely indicate a carbonate-derived component, which would naturally contribute Ca to the fluid phase during carbonate breakdown. Regarding Na, the NaCl content and salinity of the fluid were not directly measured in this study but were estimated using a salinity value of 15 wt.% NaCl. This value lies within the range (12–22 wt.% NaCl) determined in our previous fluid inclusion study from the same rock (Sumdo eclogite, Liu et al., 2019). The use of the average value allows consistent comparison among different inclusions analyzed in this study. Therefore, the

inference that the fluids were Na-rich remains robust. Following the reviewer's suggestion, we have clarified in the revised manuscript that the absolute concentrations of Ca and Na may contain minor uncertainties due to potential host–fluid interaction and the estimation of salinity. However, these uncertainties do not affect the overall interpretation that the fluids are solute-rich and capable of transporting high sulfur concentrations.

Relevant clarifications have now been added to the revised manuscript (Lines 279-282) and read as follows: “In addition, we acknowledge that the absolute concentrations of calcium and sodium may be slightly influenced by minor exchange with the host omphacite and by uncertainties associated with estimating fluid salinity using an average value. However, such effects do not alter the overall characterization of the fluids as being enriched in calcium and sodium.”

Line311-312: delete "capable of transporting high solute loads (solute-rich already implies this)
Reply: Ok, we have deleted it.

Line371-373: How does the difference between isotope-based models of fluid fraction (5%) reinforces the robustness of your estimate of 0.2-0.6%? To me this looks like a 10x discrepancy. Maybe delete “thereby reinforcing the robustness of our mass-balance constraints”
Reply: Ok, we have deleted it.

Line545: You state that you calculated “ the global H₂O flux from sub-arc depth”, but to me it seems you just quantify H₂O from the AOC, based on your previous sentence – isn't this a global flux estimate for the AOC only, not including sediments or serpentinites?
Reply: We thank the reviewer for pointing this out. We have clarified it in the revised manuscript (Lines 532-533), and the text now reads: “in order to quantify the global H₂O flux released from altered oceanic crust at sub-arc depths (~80–90 km).”

We sincerely thank Dr. Zoltán Taracsák again for the insightful and detailed comments and suggestions, which have helped improve the clarity and quality of the manuscript.

Reviewer 2 Prof. Lifei Zhang

I just have checked this revised manuscript carefully. All my questions and suggestions have been accepted and changed in the revised version. I suggest to accepted for the publication in NC.

Reply: We sincerely thank Prof. Lifei Zhang for the positive evaluation of our work and for the constructive comments provided during the previous review round, which have significantly improved the quality of the manuscript.

References

Liu, H., Xiao, Y., van den Kerkhof, A., Wang, Y., Zeng, L. & Guo, H. Metamorphism and fluid evolution of the Sumdo eclogite, Tibet: constraints from mineral chemistry, fluid inclusions and oxygen isotopes. *J. Asian Earth Sci.* 172, 292–307 (2019).